# Improving Visual Discriminability of CLIP for Training-Free Open-Vocabulary Semantic Segmentation

**Jinxin Zhou**                                                                      *zhou.3820@osu.edu*
*Department of Computer Science*
*The Ohio State University*

**Jiachen Jiang**                                                                    *jiang.2880@osu.edu*
*Department of Computer Science*
*The Ohio State University*

**Zhihui Zhu**                                                                       *zhu.3440@osu.edu*
*Department of Computer Science*
*The Ohio State University*

**Reviewed on OpenReview:** *https://openreview.net/forum?id=9spNW3DXg5*

## Abstract

Extending CLIP models to semantic segmentation remains challenging due to the misalignment between their image-level pre-training objectives and the pixel-level visual understanding required for dense prediction. While prior efforts have achieved encouraging results by reorganizing the final layer and features, they often inherit the global alignment bias of preceding layers, leading to suboptimal segmentation performance. In this work, we propose LHT-CLIP, a novel training-free framework that systematically exploits the visual discriminability of CLIP across *layer*, *head*, and *token* levels. Through comprehensive analysis, we reveal three key insights: (i) the final layers primarily strengthen image–text alignment with sacrifice of visual discriminability (e.g., last 3 layers in ViT-B/16 and 8 layers in ViT-L/14), partly due to the emergence of anomalous tokens; (ii) a subset of attention heads (e.g., 10 out of 144 in ViT-B/16) display consistently strong visual discriminability across datasets; (iii) abnormal tokens display sparse and consistent activation pattern compared to normal tokens. Based on these findings, we propose three complementary techniques: semantic-spatial reweighting, selective head enhancement, and abnormal token replacement to effectively restore visual discriminability and improve segmentation performance without any additional training, auxiliary pre-trained networks, or extensive hyperparameter tuning. Comprehensive experiments on eight widely used semantic segmentation benchmarks demonstrate that LHT-CLIP achieves substantial performance improvements across diverse scenarios, underscoring its effectiveness and practicality for real-world deployment. The source code is available at `https://jinxinzhou.github.io/LTH-CLIP/`

## 1 Introduction

Recent advances in vision-language pretrained models, such as CLIP (Radford et al., 2021), have demonstrated remarkable generalization and open-vocabulary recognition capabilities at the image level, thereby opening up possibilities for transferring image-text alignment to pixel-level tasks. Despite this progress, they often underperform in dense prediction tasks like semantic segmentation, primarily due to their limited capacity to localize fine-grained visual details (Rao et al., 2022; Wang et al., 2024). To address these limitations, several studies have incorporated trainable modules into CLIP, typically relying on additional forms of supervision such as dense annotations for a restricted set of categories (Xu et al., 2022b;a; Xing et al., 2023; Cho et al., 2024; Li et al., 2024a;b) or supplementary image-text pairs (Cha et al., 2023; Luo et al.,

2023a; Ren et al., 2023; Xu et al., 2023b; Zhang et al., 2023). Although these approaches have demonstrated improved segmentation performance, they incur significant computational and annotation costs. Furthermore, the dependence on limited supervision undermines the generalizability of the model, making it prone to overfitting the training distribution.

These challenges have sparked increasing interest in training-free methods (Wang et al., 2024; Li et al., 2023; Zhou et al., 2022a; Lan et al., 2024a; Hajimiri et al., 2025; Lan et al., 2024b; Shao et al., 2024; Bousselham et al., 2024; Yang et al., 2024), which aim to adapt CLIP's pre-trained representations for semantic segmentation without additional training, while preserving its generalization capability. A key difficulty in this direction is enhancing visual representations for accurate pixel-level predictions. For instance, MaskCLIP (Zhou et al., 2022a) computes similarity between key features in the final attention layer to enrich patch embeddings. SCLIP (Wang et al., 2024) replaces the standard query-key attention with correlative self-attention (query-query and key-key). ClearCLIP (Lan et al., 2024a) further removes residual connections and discards the FFN in the final layer to reduce noise and improve spatial alignment. ResCLIP (Yang et al., 2024) incorporates attention maps from earlier layers to refine final-layer attention map. However, these methods largely focus on modifying the final-layer attention, often leading to suboptimal ambiguous local relationships and noisy segmentation. To address spatial limitations, some approaches incorporate features from auxiliary backbones such as DINO (Wysoczańska et al., 2024; Lan et al., 2024b), SAM (Lan et al., 2024b; Zhang et al., 2025; Shi et al., 2025), or diffusion models (Corradini et al., 2024; Sun et al., 2024; Zhang et al., 2025). While effective, these methods incur significant computational and memory overhead.

Motivated by these limitations, we begin with a layer-wise analysis of visual discriminability and text-semantic alignment within the CLIP vision encoder (see Section 2.2 for detailed definitions). As shown in Figure 1, the final layers exhibit a clear trade-off: visual discriminability drops sharply while semantic alignment improves only marginally. To understand the cause of this phenomenon, we further examine internal token interactions and structural patterns across layers. Attention map visualizations reveal that abnormal tokens emerge in deeper layers, attracting disproportionately high attention from nearly all spatial positions. This behavior causes the majority of tokens to converge on a small subset, thus disrupting the spatial coherence. Further analysis reveals that these abnormal tokens have sparse, high-magnitude activations that remain consistent across positions, layers, and samples. Complementary to prior assumptions that such tokens primally encode global semantic content, our findings suggest they act more as bias components offsetting global mean features, thereby facilitating alignment with text embeddings.

Based on the analysis, we propose LHT-CLIP, a training-free framework that leverages the inherent properties of CLIP to enhance the visual discriminability while preserving semantic alignment. LHT-CLIP comprises three complementary strategies: abnormal token replacement (ATR), spatial-semantic reweighting (SSR), and selective head enhancement (SHE). Specifically, ATR identifies abnormal tokens via sparsity thresholding and replaces them with neighboring tokens. SSR mitigates the degradation of visual discriminability in the final layers by upweighting residual pathways, thereby restoring balance between spatial coherence and semantic alignment. Finally, SHE further enhances visual discriminability by selectively aggregating features from high-discriminability attention heads, using them as soft pseudo-masks to refine output features. Experimental results show that LHT-CLIP consistently improves performance when integrated into diverse baselines, achieving significant improved results on eight benchmark datasets.

**Comparison with previous work.** While prior work (Shao et al., 2024; Bai et al., 2024) has similarly identified the presence of anomalous tokens, our study provides a more comprehensive characterization of their distinguishing properties, demonstrating that these tokens elicit sparse, high-magnitude activations that remain consistent across positions, layers, and samples. Moreover, existing approaches rely on sophisticated anomaly-detection techniques and require careful tuning of multiple hyperparameters, such as the neighborhood radius and minimum cluster size in DBSCAN (Ester et al., 1996) as used in Shao et al. (2024), and the neighborhood size and contamination level in LOF (Breunig et al., 2000) as used in Bai et al. (2024). In contrast, our ATR method directly leverages the consistent property of sparsity in anomalous tokens, thereby reducing the need for intricate hyperparameter selection and showing hyperparameter robustness across diverse datasets, as shown in Table 2. Furthermore, to improve open-vocabulary semantic segmentation performance, prior work has explored leveraging earlier layers, either by reorganizing the final-layer

computation using attention maps derived from preceding layers (Yang et al., 2024) or by designing multi-layer fusion strategies (Yang et al., 2024; Bai et al., 2024). However, these approaches leave the underlying inference behavior of earlier layers unchanged, which can lead to suboptimal results due to the degraded visual discriminability typically exhibited by intermediate representations. In contrast, our proposed SSR strategy explicitly modifies the inference procedure by upweighting the residual components, thereby substantially enhancing the visual discriminability of intermediate features, as illustrated in Figure 5. Moreover, SSR is orthogonal to these prior methods and can be naturally integrated with them.

**Contributions.** Our contributions can be summarized as follows:

- We conduct a throughout analysis of visual discriminability at the token, head, and layer levels.

- We propose, a novel training-free approach, terms LHT-CLIP. To the best of our knowledge, this is the first work to explicitly modify the inference procedure prior to the final layer, enabling improved spatial coherence without compromising semantic alignment.

- The extensive experiment results on open-vocabulary semantic segmentation tasks consistently demonstrate the effectiveness of the proposed method.

## 2 Analysis of Visual Discriminability and Semantic Alignment

### 2.1 Preliminaries

CLIP employs a Vision Transformer (ViT) (Dosovitskiy et al., 2020) as its image encoder to generate visual representations that are aligned with corresponding textual descriptions. The vision encoder first tokenizes an input image of size $H \times W \times 3$ by dividing it into a grid of non-overlapping patches of size $P \times P$, yielding $h = H/P$ rows and $w = W/P$ columns of patches. Each patch is then linearly projected into a $D$-dimensional embedding space, $\boldsymbol{x}_i \in \mathbb{R}^D$, and augmented with positional embeddings. An additional learnable `[CLS]` token is prepended to the sequence and is later used for image-level prediction. The resulting token sequence is denoted as $\boldsymbol{X}^0 = [\boldsymbol{x}^0_{\text{cls}}, \boldsymbol{x}^0_1, \ldots, \boldsymbol{x}^0_{hw}] \in \mathbb{R}^{(1+hw) \times D}$. This sequence is passed through a stack of $L$ Transformer encoder layers, each consisting of a multi-head self-attention (MSA) module followed by a feed-forward network (FFN). Let $\text{LN}(\cdot)$ denotes layer normalization, the token representations are updated at each layer $l$ as follow:

$$\hat{\boldsymbol{X}}^l = \boldsymbol{X}^{l-1} + \text{MSA}(\text{LN}(\boldsymbol{X}^{l-1})), \tag{1}$$

$$\boldsymbol{X}^l = \hat{\boldsymbol{X}}^l + \text{FFN}(\text{LN}(\hat{\boldsymbol{X}}^l)). \tag{2}$$

The CLIP model is originally trained on large-scale image–text pairs for open-vocabulary image recognition tasks. To extend it to semantic segmentation, a natural approach is to compute the similarity between the visual tokens $\boldsymbol{X}^L = [\boldsymbol{x}^L_1, \ldots, \boldsymbol{x}^L_{hw}]$ from the final Transformer layer and the textual embeddings of $C$ category names, denoted by $\mathbf{t} \in \mathbb{R}^{C \times D}$. This results in a patch-text similarity map of size $hw \times C$. Denote $\mathbf{t}_c$ as the embedding of the $c$-th class name, the final segmentation prediction is obtained by applying an `argmax` operation over the class dimension of this similarity map, as follows:

$$\hat{c}(\boldsymbol{x}_i) = \arg\max_c \frac{\langle \boldsymbol{x}^L_i, \mathbf{t}_c \rangle}{\|\boldsymbol{x}^L_i\| \cdot \|\mathbf{t}_c\|}. \tag{3}$$

Ideally, for effective semantic segmentation, the vision encoder should produce feature representations that satisfy two key properties:

- **Visual discriminability**: token features should exhibit high internal consistency within the same semantic category while remaining clearly distinguishable from those of other categories, thereby enabling accurate and clean segmentation results.

- **Semantic alignment**: token features should be well-aligned with their corresponding textual embeddings to enable semantically meaningful segmentation results.

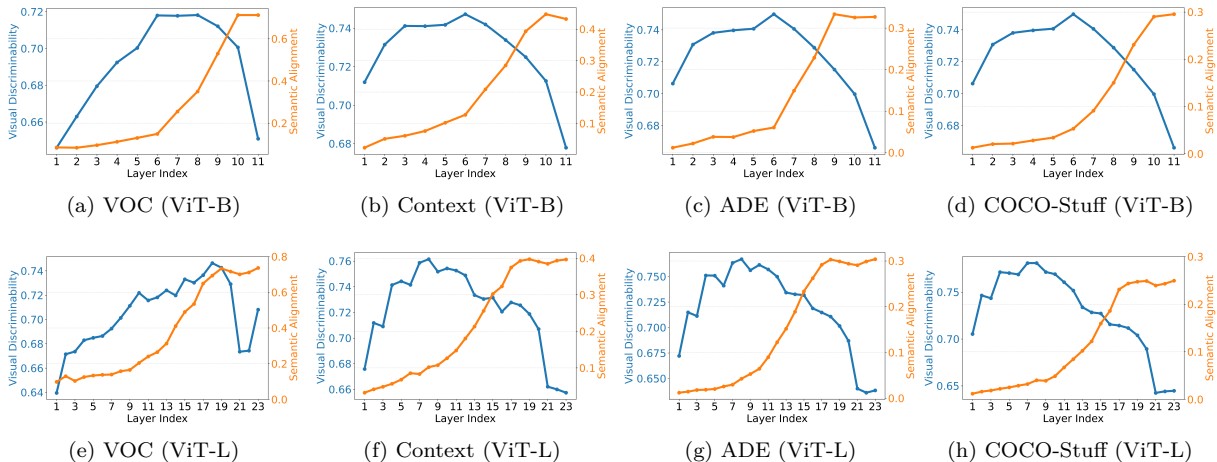

Figure 1: Layer-wise analysis of visual discriminability (blue) and semantic alignment (orange) within the CLIP vision encoders across different datasets. The final layer is excluded from the analysis to avoid discrepancies caused by prior modifications to the last-layer in different methods.

Beyond their importance in open-vocabulary semantic segmentation, these two properties are also highly relevant to the development of multimodal large language models. To preserve strong generalization capability, the vision encoder of CLIP is often directly employed to extract visual representations without additional training, which are then used as inputs to downstream language models, as exemplified by LLaVA (Liu et al., 2023; 2024).

## 2.2 Analysis of visual discriminability and semantic alignment

**Measures of visual discriminability and semantic alignment.** To quantify **visual discriminability**, we adopt the evaluation protocol proposed by Mukhoti et al. (2023). Specifically, let $\mathbf{x}_i^l \in \mathbb{R}^D$ and $\mathbf{x}_j^l \in \mathbb{R}^D$ denote the features of two image patches $i$ and $j$ extracted from the $l$-th layer. Each feature vector is $\ell_2$-normalized, and their cosine similarity is computed to serve as the prediction of a binary classifier that determines whether the two patches belong to the same semantic category. Given the corresponding semantic labels $t(\boldsymbol{x}_i)$ and $t(\boldsymbol{x}_j)$, the target label for classification is set to 1 if $t(\boldsymbol{x}_i) = t(\boldsymbol{x}_j)$, and 0 otherwise. Performance on this binary classification task provides a measure of visual discriminability, since effective representations should yield high cosine similarity for patches from the same semantic class and low similarity otherwise. To evaluate **semantic alignment**, we extract the intermediate representations $\mathbf{x}_i^l \in \mathbb{R}^D$ from each individual visual token at layer $l$, and project them into the final visual–text aligned space using the last ViT layer. Based on these projected features, semantic alignment is quantified as the average accuracy between predicted and ground-truth semantic labels, following Equation (3). To prevent contamination from extraneous semantic information and noisy integration during the final attention computation, we follow prior work (Lan et al., 2024a; Zhou et al., 2022a) by replacing the attention matrix with an identity matrix and removing both the FFN and residual connections of the last ViT layer.

**Sharp decline in visual discriminability with marginal gains in semantic alignment in the final layers.** To analyze the layer-wise dynamics of visual discriminability and semantic alignment, we investigate on four datasets: Pascal VOC (Everingham & Winn, 2011), PASCAL Context (Mottaghi et al., 2014), ADE20K (Zhou et al., 2017), and COCO-Stuff (Caesar et al., 2018). For each dataset, we evaluate the ViT-B/16 and ViT-L/14 variants of the CLIP vision encoder using 1,000 randomly selected training samples. As shown in Figure 1, we observe a consistent pattern across datasets:

- *Visual discriminability follows an inverted U-shaped curve across layers*: it increases in the early stages but declines sharply in the deeper layers. For instance, in the ViT-B/16 model, the last two layers preceding

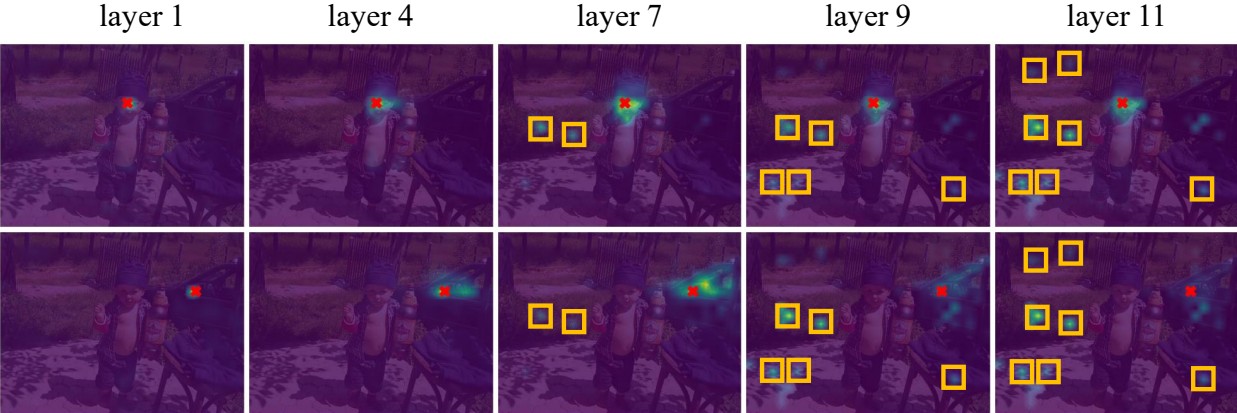

Figure 2: Abnormal token phenomenon in attention maps across different layers of the ViT-B/16 model used as the CLIP vision encoder. Attention maps are computed with respect to specific visual token positions, denoted by × (e.g., the "child" token in the top row and the "car" token in the bottom row). Representative abnormal tokens are highlighted with orange boxes.

the final layer exhibit a pronounced reduction in visual discriminability, while in the ViT-L/14 model, a similar decline is observed over the last seven layers.

- *Semantic alignment exhibits an approximately monotonic increase across layers*: it improves substantially in the early stages but gradually saturates in the later layers, yielding only marginal gains thereafter.

These observations offer a nuanced understanding of why CLIP has proven effective for open-vocabulary semantic segmentation. In particular, the strong semantic alignment observed in the final layers explains why prior work can leverage last-layer features for aligning visual tokens with textual categories. However, the significant decline in visual discriminability in last layers reveals a key limitation as they may lack the fine-grained visual distinctions necessary for producing accurate and precise segmentation masks. In this work, we aim to mitigate this limitation by proposing methods that jointly improve visual discriminability and preserve semantic alignment. Before introducing our approach, we first investigate the underlying causes of the decline in visual discriminability.

**Emergence of abnormal tokens with high norms and sparse activations.** To understand the cause of the sharp decline observed in the final layers, we analyze the attention maps among visual tokens across different layers. As shown in Figure 2, deeper layers exhibit a small set of dominant tokens that receive disproportionately high attention from nearly all visual tokens, causing most tokens to focus on this subset, consistent with prior observations (Darcet et al., 2023; Shao et al., 2024). The presence of dominant tokens attracts visual tokens to become similar, thereby reducing visual discriminability and degrading segmentation result. To further characterize these dominant tokens, we compare their features with those of normal tokens. As illustrated in Figure 3, dominant tokens exhibit sparse activation patterns, with only a few channels maintaining high activation. To quantify this sparsity, we adopt the hoyer score (Hoyer, 2004):

$$\mathcal{H}(\mathbf{x}_i^l) = \frac{\sqrt{D} - \frac{\|\mathbf{x}_i^l\|_1}{\|\mathbf{x}_i^l\|_2}}{\sqrt{D} - 1} \in [0, 1]. \tag{4}$$

Let $\mathbf{x}_i^l \in \mathbb{R}^D$ denote the feature vector of the $i$-th token at layer $l$. A higher score corresponds to sparser activations. We employ this metric to quantify sparsity and to visualize its distribution across layers and token positions in Figure 3(d). As shown in the figure, sparse high-norm abnormal tokens begin to emerge in the middle layers and persist at the same positions in subsequent layers.

**Abnormal tokens encode global information while being dominated by a bias-like component.** To further explore the information encoded in abnormal tokens, we analyze their pairwise cosine similarity

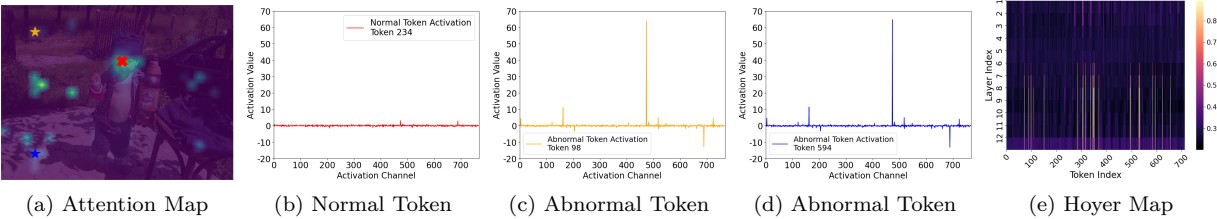

(a) Attention Map    (b) Normal Token    (c) Abnormal Token    (d) Abnormal Token    (e) Hoyer Map

Figure 3: Illustration of the sparsity and high-norm characteristics of abnormal tokens. Figure (a) shows the attention map of the red anchor token ✕. Figures (b)–(d) depict the channel activations of a normal token (red ✕) and two abnormal tokens (orange ★ and blue ★) highlighted in Figure (a). Figure (e) presents the hoyer score distribution across layers and token positions.

with normal tokens and the [CLS] token on the same images from the ImageNet (Deng et al., 2009) validation set. As shown in Figure 4(a), the cosine similarity with [CLS] increases while the similarity with normal tokens decreases as the layer depth increases, suggesting that abnormal tokens progressively align with global information and discard local details. Interestingly, when analyzing their pairwise cosine similarity across different positions, layers, and data samples, we find that these tokens exhibit consistently high similarity, with cosine values exceeding 0.98, as shown in Figure 4. Complementary to prior discovery, this indicates that abnormal tokens primarily act as bias components that offset global mean features, thereby facilitating text alignment in a manner analogous to the bias term in final-layer classifiers under neural collapse (Zhu et al., 2021; Zhou et al., 2022b). However, both global information and bias-like components in abnormal tokens are detrimental for segmentation tasks, which rely on fine-grained local understanding.

**A subset of attention heads exhibits consistently strong visual discriminability.** To enhance the visual discriminability of last-layer features, a natural strategy is to exploit the more discriminative intermediate features to construct a pseudo mask, guiding the reorganization of the final-layer features, as indicated by the analysis in Figure 1. Inspired by recent studies (Gandelsman et al., 2023; Kang et al., 2025) showing that different attention heads capture distinct visual concepts, such as number, shape and texture, we take a step further by investigating whether specific heads are particularly responsible for encoding visual discriminability. To identify such heads, we follow the formulation introduced in Elhage et al. (2021); Gandelsman et al. (2023), which rewrites the MSA output as a summation over $H$ independent attention heads: $\text{MSA}(\text{LN}(\boldsymbol{X}^l)) = \sum_{h=1}^{H} \mathbf{A}_h^l \mathbf{V}_h^l \mathbf{W}_o^l \in \mathbb{R}^{(1+hw) \times D}$, where $\mathbf{A}_h^l$ and $\mathbf{V}_h^l$ denote the attention and value matrices for the $h$-th head at layer $l$, and $\mathbf{W}_o^l$ is the output projection matrix shared across all heads. Accordingly, the features of the $h$-th head at layer $l$ are expressed as:

$$\boldsymbol{X}^{l,h} = \mathbf{A}_h^l \mathbf{V}_h^l \mathbf{W}_o^l. \tag{5}$$

In Figure 6, we show the distribution of visual discriminability across attention heads for ViT-B/16. From the figure, we observe that the output features of certain attention heads, such as the 11th head in the 6th layer, consistently exhibit high visual discriminability across different datasets, suggesting that there are a subset of heads which are more effective in capturing locally distinguishable features.

## 3 Method for Improving Visual Discriminability

In this section, we introduce our training-free framework, which comprises three components: Abnormal Token Replacement (ATR) in Section 3.1, Spatial-Semantic Reweighting (SSR) in Section 3.2, and Selective Head Enhancement (SHE) in Section 3.3. Each component is complementary, and together they work synergistically to enhance the visual discriminability of the CLIP model.

### 3.1 Abnormal token replacement (ATR)

**Identify and replace abnormal tokens with their neighbors.** To mitigate the adverse effects of anomalous tokens, we propose a simple and effective strategy to suppress their impact. As shown in earlier

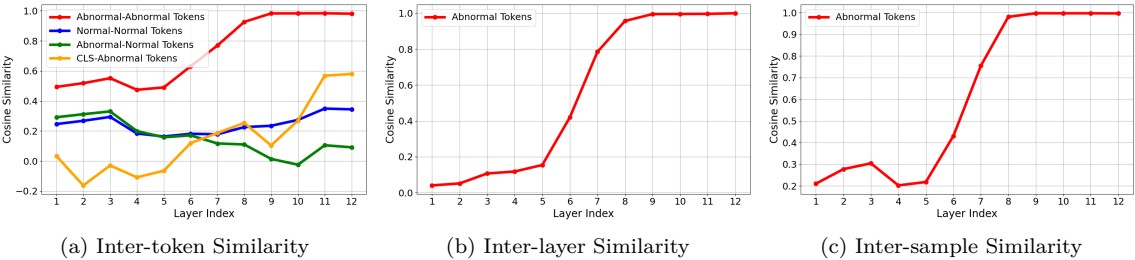

(a) Inter-token Similarity      (b) Inter-layer Similarity      (c) Inter-sample Similarity

Figure 4: Layer-wise cosine similarity among abnormal tokens across positions, layers and samples.

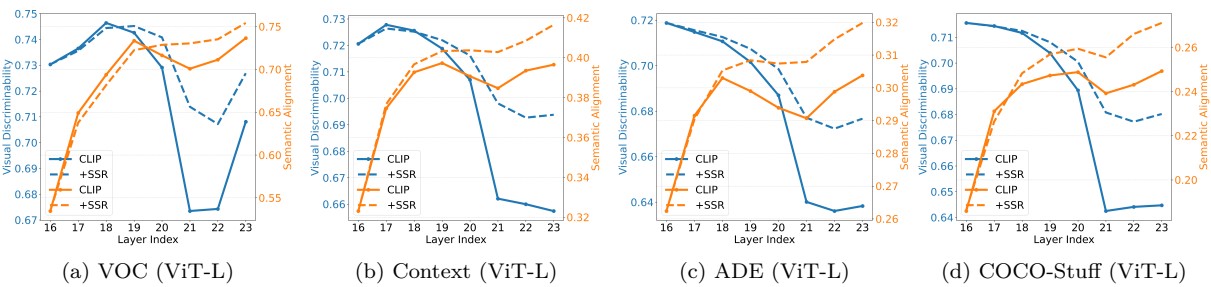

(a) VOC (ViT-L)    (b) Context (ViT-L)    (c) ADE (ViT-L)    (d) COCO-Stuff (ViT-L)

Figure 5: Layer-wise visual discriminability (blue) and semantic alignment (orange) for ViT-L/14 on four benchmarks. Solid lines denote the baseline CLIP (Raw) performance, while dashed lines indicate results after applying the proposed SSR strategy. The application of SSR notably improves visual discriminability in final layers and consistently enhances semantic alignment across all datasets.

analysis, these tokens are characterized by high norms and sparse activations. To identify them systematically, we compare a norm-based and a sparsity-based criterion. Empirically, we find that the sparsity-based approach is more robust to hyperparameter selection (see Section A.3.3); therefore, we adopt the hoyer score $\mathcal{H}(\mathbf{x}_i^l)$ defined before as a sparsity-based criterion. Tokens with scores exceeding a predefined threshold $\tau$ are deemed anomalous and grouped into the set $\mathcal{A}_l = \{i | \mathcal{H}(\mathbf{x}_i^l) > \tau\}$. To mitigate their impact, each anomalous token at spatial position $(m, n) \in \mathcal{A}$ is substituted with a weighted aggregation of its eight nearest neighbors:

$$\boldsymbol{x}_{m,n}^l = \frac{\sum_{i=m-1}^{m+1} \sum_{j=n-1}^{n+1} \mathbb{1}((i,j) \notin \mathcal{A}) \boldsymbol{x}_{i,j}^l}{\sum_{i=m-1}^{m+1} \sum_{j=n-1}^{n+1} \mathbb{1}((i,j) \notin \mathcal{A})}, \quad \forall (m,n) \in \mathcal{A}. \tag{6}$$

Here, $\mathbb{1}(\cdot)$ ensure that only normal tokens contribute to the replacement of anomalous ones. Empirically, we find that applying this strategy before the penultimate layer leads to a performance drop, likely due to the removal of inherent biases encoded in abnormal tokens, which substantially alters the inference process. Therefore, we apply it only at the penultimate layer, i.e., with $l = L - 1$.

## 3.2 Spatial-semantic reweighting (SSR)

**Upweight residual component to improve visual discriminability.** After mitigating the impact of anomalous tokens at the penultimate layer, the final features exhibit improved visual discriminability (see Figure 8 in Section A.2.2). However, a critical challenge remains: anomalous tokens appeared in earlier layers already degrade visual discriminability, limiting the effectiveness of final-layer refinements. To address it, we propose a spatial–semantic reweighting strategy, based on our layer-wise analysis showing that the final layers provide only marginal gains in semantic alignment while substantially reducing visual discriminability. Overall, the proposed strategy aims to enhance the model's spatial coherence while preserving its semantic alignment. Specifically, given the feature representation $\boldsymbol{X}^{l-1}$ at the $l$-th layer within the final few layers (e.g., layers 10–11 in ViT-B/16 and layers 17–23 in ViT-L/14), we reweight the forward pass by upweighting

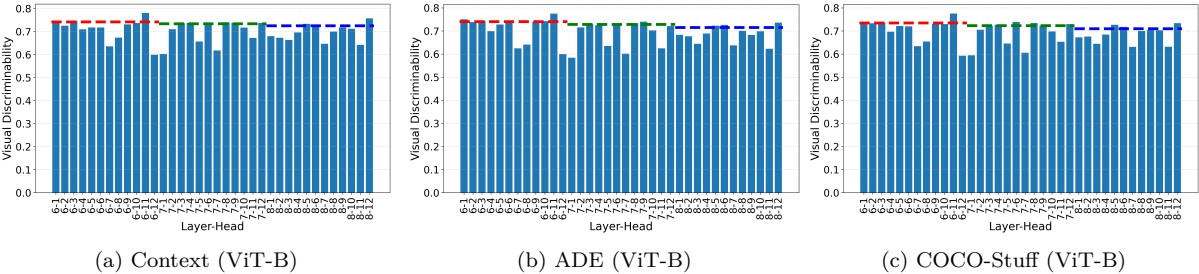

(a) Context (ViT-B)         (b) ADE (ViT-B)         (c) COCO-Stuff (ViT-B)

Figure 6: Head-wise visual discriminability analysis across multiple datasets using the ViT-B/16 backbone. The dashed lines in different colors denote the corresponding layer-wise visual discriminability scores. For clarity, heads from three layers (i.e., the 6th, 7th, and 8th layers) are displayed.

the residual pathway and downweighting the attention and MLP submodules, as follows:

$$\hat{\boldsymbol{X}}^l = (1 + \alpha)\boldsymbol{X}^{l-1} + (1 - \alpha)\mathrm{MSA}(\mathrm{LN}(\boldsymbol{X}^{l-1})), \tag{7}$$

$$\boldsymbol{X}^l = (1 + \alpha)\hat{\boldsymbol{X}}^l + (1 - \alpha)\mathrm{FFN}(\mathrm{LN}(\hat{\boldsymbol{X}}^l)), \tag{8}$$

where $\alpha \in [0, 1]$ is a reweighting coefficient that controls the emphasis on the residual signal. As $\alpha$ increases, the $l$-th block preserves more visually discriminative features from earlier layers through the residual pathway, while reducing the adverse impact of noisy semantic aggregation in the MSA and FFN submodules. To the best of our knowledge, prior work has primarily focused on reforming the final layer or modifying its representations to improve performance. However, such approaches inevitably inherit the global semantic alignment bias of the proceeding layers, leading to suboptimal segmentation due to substantially reduced visual discriminability. In contrast, our SSR strategy explicitly addresses this limitation by rebalancing residual and semantic contributions, thereby improving the visual discriminability of intermediate features.

**Effectiveness of SSR in enhancing visual discriminability of preceding layers.** Since our SSR strategy is applied not only to the penultimate layer but also to earlier layers, we further present the layer-wise curves of visual discriminability and semantic alignment after applying SSR in Figure 5. As illustrated in the figure, the visual discriminability of the final few layers is significantly enhanced, demonstrating the effectiveness of the proposed SSR strategy for visual discriminability improvement. Moreover, semantic alignment also exhibits consistent improvements, as improved visual discriminability reduces spurious semantic aggregation from other noisy tokens.

### 3.3 Selective head enhancement (SHE)

**Construct pseudo masks from discriminative heads to further enhance penultimate features.** Based on the head-wise analysis, we leverage high-performing heads to construct soft pseudo masks, which are used to enhance the visual discriminability of the penultimate features. Specifically, let $\mathrm{VD}_{l,h}^s$ denote the visual discriminability score of the $h$-th head in the $l$-th layer for dataset $s \in \{\mathrm{Context}, \mathrm{ADE}, \mathrm{Stuff}\}$. To obtain a dataset-agnostic measurement, we compute the average score (described in Section 2.2) across datasets, denoted as $\overline{\mathrm{VD}}_{l,h}$. We then rank heads by their $\overline{\mathrm{VD}}_{l,h}$ values (see Figure 9 and Figure 10) and select the top-$k$ heads to form the set $\mathcal{H}_k$. The corresponding features are aggregated as $\overline{\boldsymbol{X}}_k = \frac{1}{k}\sum_{(l,h)\in\mathcal{H}_k}\boldsymbol{X}^{l,h}$, and a similarity map is constructed as $S = \overline{\boldsymbol{X}}_k\overline{\boldsymbol{X}}_k^\top / \|\overline{\boldsymbol{X}}_k\|^2$ to capture the pairwise similarity among visual tokens. To suppress the spurious interactions between tokens from different semantic categories, we apply a thresholding operation with a parameter $\beta$, yielding the filtered similarity map $S_\beta$, where $S_\beta(i,j) = S(i,j)$ if $S(i,j) \geq \beta$, and $S_\beta(i,j) = 0$ otherwise. This filtered similarity map $S_\beta$ serves as a soft pseudo-mask, which is column-normalized and used to refine the final-layer features as $\boldsymbol{X}^{L-1} \leftarrow \mathrm{Norm}(S_\beta)\boldsymbol{X}^{L-1}$.

# 4 Experiment Results

**Evaluation Protocol.** We follow the evaluation protocol from prior works (Wang et al., 2024; Lan et al., 2024a; Hajimiri et al., 2025) and assess our method on eight widely used semantic segmentation benchmarks. We group them into two categories and use abbreviated names for clarity. The first category excludes background and includes Pascal VOC (Everingham & Winn, 2011) (VOC20), Pascal Context (Mottaghi et al., 2014) (C59), COCO-Stuff (Caesar et al., 2018) (Stuff), ADE20K (Zhou et al., 2017) (ADE), and Cityscapes (Cordts et al., 2016) (City). The second includes background and consists of VOC21, C60, and COCO-Object (Caesar et al., 2018) (Object). We use ViT-B/16 and ViT-L/14 as the vision encoder of CLIP (Radford et al., 2021), and report results using the mean Intersection-over-Union (mIoU). All hyper-parameters are tuned on 1,000 randomly sampled training images from {Context, ADE, Stuff } datasets and kept fixed without task-specific tuning during evaluation, based on ablation study in Section 4.2. Additional details are provided in the appendix.

## 4.1 Comparison with existing methods.

We compare our approach against a comprehensive set of open-vocabulary semantic segmentation methods, including the direct baseline CLIP (Radford et al., 2021), as well as several state-of-the-art training-free approaches: MaskCLIP (Zhou et al., 2022a), CLIPSurgery (Li et al., 2023), SCLIP (Wang et al., 2024), NA-CLIP (Hajimiri et al., 2025), ClearCLIP (Lan et al., 2024a), LAVG (Kang & Cho, 2024), and ResCLIP (Yang et al., 2024). We also include several influential weakly supervised methods, such as GroupViT (Xu et al., 2022a), ReCo (Shin et al., 2022), and TCL (Cha et al., 2023). Unless otherwise specified, all reported results are taken directly from the respective original papers and ResCLIP (Yang et al., 2024). As our method is orthogonal to approaches that primarily target improvements in the final-layer attention, we evaluate its effectiveness when integrated with recent state-of-the-art methods that employ specialized attention mechanisms in the last layer, including SCLIP (Wang et al., 2024), ClearCLIP (Lan et al., 2024a), and ResCLIP (Yang et al., 2024). For fair comparison, we exclude the *Semantic Feedback Refinement* module in ResCLIP, as it relies on the computationally expensive PAMR (Araslanov & Roth, 2020) post-processing, which is inconsistent with our evaluation setting.

In Table 1, we summarize the performance of various previous methods on benchmark datasets using the ViT-B/16 backbone. Our proposed LHT-CLIP consistently enhances the performance of state-of-the-art approaches, including SCLIP (Wang et al., 2024), ClearCLIP (Lan et al., 2024a), and ResCLIP (Yang et al., 2024). Notably, when integrated with ResCLIP (Yang et al., 2024), LHT-CLIP achieves significantly improved results, outperforming leading weakly supervised methods. As a plug-and-play solution, LHT-CLIP yields consistent improvements across all datasets compared to the respective baselines, demonstrating its strong generalization capability. For comprehensiveness, results on the ViT-L/14 backbone are provided in Table 8. In line with observations from Yang et al. (2024), existing methods generally exhibit a performance drop exceeding 2% mIoU when adapting to a different backbone; for instance, ClearCLIP (Lan et al., 2024a) suffers a notable decline of 2.7% mIoU. In contrast, when augmented with LHT-CLIP, this performance degradation is significantly alleviated, highlighting the robustness of our approach. Across both backbones, LHT-CLIP delivers substantial improvements over baseline methods, validating its effectiveness.

## 4.2 Experimental analysis

In this section, we conduct comprehensive ablation studies to validate the effectiveness of our proposed method. We adopt SCLIP (Wang et al., 2024) as the baseline, which enhances spatial correlation by modifying the attention mechanism in the final layer, replacing the standard $QK^\top$ attention with a combination of $QQ^\top + KK^\top$. In addition, following prior work (Lan et al., 2024a; Yang et al., 2024), we remove the residual connections and FFN from the final transformer layer. For experiments in this part, we highlight the optimal hyperparameter settings in gray .

**Analysis of the hoyer threshold parameter $\tau$.** Our method relies on hoyer sparsity to identify anomalous tokens, making the sparsity threshold $\tau$ a critical hyperparameter. In Table 2, We conduct a systematic evaluation. At $\tau = 0.2$, many normal tokens are misclassified, leading to excessive smoothing and degraded

Table 1: Performance comparison of our approach with other methods on eight semantic segmentation benchmarks. Existing methods with our improvement are marked in  gray .

| Methods | Training | With a background class | | | Without background class | | | | | Avg. |
|---|---|---|---|---|---|---|---|---|---|---|
| | | VOC21 | C60 | Object | VOC20 | City | C59 | ADE | Stuff | |
| ReCo | ✓ | 25.1 | 19.9 | 15.7 | 57.7 | 21.1 | 22.3 | 11.2 | 14.8 | 23.5 |
| GroupViT | ✓ | 52.3 | 18.7 | 27.5 | 79.7 | 18.5 | 23.4 | 10.4 | 15.3 | 30.7 |
| TCL | ✓ | 51.2 | 24.3 | 30.4 | 77.5 | 23.1 | 30.3 | 14.9 | 19.6 | 33.9 |
| CLIP | ✗ | 16.2 | 7.7 | 5.5 | 41.8 | 5.5 | 9.2 | 2.1 | 4.4 | 11.6 |
| MaskCLIP | ✗ | 38.8 | 23.6 | 20.6 | 74.9 | 16.4 | 26.4 | 9.8 | 14.8 | 28.2 |
| CLIPSurgery | ✗ | 55.2 | 18.7 | 27.5 | 79.7 | 18.5 | 23.4 | 10.4 | 15.3 | 31.1 |
| LaVG | ✗ | 62.1 | 31.6 | 34.2 | 82.5 | 26.2 | 34.7 | 15.8 | 23.2 | 38.8 |
| NACLIP | ✗ | 58.9 | 32.2 | 33.2 | 79.7 | 35.5 | 35.2 | 17.4 | 23.3 | 39.4 |
| SCLIP | ✗ | 59.7 | 31.7 | 33.5 | 81.5 | 32.3 | 34.5 | 16.5 | 22.7 | 39.1 |
| +LHT-CLIP (ours) | ✗ | 64.8 | 34.8 | 36.6 | 86.3 | 36.1 | 37.6 | 18.0 | 24.9 | 42.4 (+3.3) |
| ClearCLIP | ✗ | 57.0 | 32.2 | 32.5 | 82.3 | 32.8 | 35.8 | 17.3 | 24.0 | 39.2 |
| +LHT-CLIP (ours) | ✗ | 63.8 | 35.2 | 35.6 | 85.7 | 37.8 | 38.8 | 19.2 | 25.8 | 42.7 (+3.5) |
| ResCLIP | ✗ | 60.0 | 32.7 | 34.0 | 85.5 | 35.6 | 35.8 | 17.7 | 23.8 | 40.6 |
| +LHT-CLIP (ours) | ✗ | 63.9 | 35.5 | 35.2 | 86.9 | 38.2 | 38.2 | 19.1 | 25.5 | 42.8 (+2.2) |

Table 2: Study of hoyer sparsity threshold $\tau$.

| $\tau$ | C60 | Stuff | C59 | ADE | Avg |
|---|---|---|---|---|---|
| $\tau = 0.2$ | 1.3 | 1.2 | 1.5 | 0.6 | 1.2 |
| $\tau = 0.4$ | 33.0 | 24.4 | 36.6 | 17.9 | 28.0 |
| $\tau = 0.5$ | 33.0 | 24.4 | 36.7 | 17.9 | 28.0 |
| $\tau = 0.8$ | 33.0 | 24.4 | 36.6 | 17.9 | 28.0 |
| $\tau = 0.9$ | 32.4 | 24.0 | 36.0 | 17.6 | 27.5 |
| baseline | 32.4 | 24.0 | 36.0 | 17.6 | 27.5 |

Table 3: Study of $(l_{\text{start}}, l_{\text{end}}, \alpha)$ in SSR module.

| $(l_{\text{start}}, l_{\text{end}}, \alpha)$ | C60 | Stuff | C59 | ADE | Avg |
|---|---|---|---|---|---|
| baseline | 32.4 | 24.0 | 36.0 | 17.6 | 27.5 |
| (9, 11, 0.1) | 32.7 | 23.7 | 36.5 | 17.7 | 27.7 |
| (10, 11, 0.1) | 33.1 | 24.3 | 36.9 | 18.0 | 28.1 |
| (11, 11, 0.1) | 32.7 | 24.4 | 36.4 | 18.0 | 27.9 |
| (10, 11, 0.05) | 32.8 | 24.3 | 36.4 | 18.0 | 27.9 |
| (10, 11, 0.2) | 32.6 | 23.9 | 36.5 | 17.8 | 27.7 |

performance. As $\tau$ increases to 0.4, performance steadily improves, but plateaus between 0.5 and 0.8, with a decline observed beyond this range. The broad stable region indicates a clear sparsity gap between normal and abnormal tokens, highlighting the robustness of ATR to threshold selection. Based on this analysis, we fix $\tau = 0.5$ for all experiments.

**Analysis of spatial-semantic reweighting parameters $\alpha$ and number of layers.** To evaluate the impact of the reweighting strength $\alpha$ and the range of layers involved, from $l_{\text{start}}$ to $l_{\text{end}}$, we perform a comprehensive sensitivity analysis. The results are summarized in Table 3. We observe that the best performance is obtained when reweighting is applied to layers 10–11 in the ViT-B/16 backbone. This aligns with our earlier findings that these layers experience a marked decline in visual discriminability while yielding only marginal improvements in semantic alignment. Extending reweighting to include layer 9 results in a slight gain in visual discriminability but introduces noisy semantic signals, ultimately leading to a reduction in segmentation performance. In addition, we examine the effect of varying the reweighting threshold parameter $\alpha$. As $\alpha$ increases from 0 to 0.1, performance improves steadily, indicating a beneficial balance between visual and semantic cues. However, further increasing $\alpha$ leads to a performance drop, as it incorporates inaccurate semantic information from earlier layers and significantly perturbs the input distribution of subsequent layers.

**Analysis of the number of selected heads $k$.** In Table 4, we study the effect of varying the number of top-$k$ attention heads selected for enhancement. For ViT-B/16 backbone, increasing $k$ from 5 to 10 improves segmentation accuracy, as aggregating multiple visual discriminative heads helps suppress spurious correlations. However, performance declines when $k$ becomes too large due to the inclusion of noisy or less informative heads, which introduce noisy cross-category interactions. We also compare head- and layer-level selection (best $l = 8$), finding that head-level selection consistently performs better, as discriminative heads are distributed across layers, while entire-layer selection introduces irrelevant heads and degrades performance.

Table 4: Study of number of selected heads $k$.

| $k$ | C60 | Stuff | C59 | ADE | Avg |
|---|---|---|---|---|---|
| baseline | 32.4 | 24.0 | 36.0 | 17.6 | 27.5 |
| layer($l = 8$) | 32.8 | 24.8 | 36.4 | 17.7 | 28.0 |
| $k = 5$ | 32.8 | 24.6 | 36.3 | 17.8 | 27.9 |
| $k = 10$ | 33.0 | 25.1 | 37.0 | 18.3 | 28.4 |
| $k = 30$ | 33.0 | 25.0 | 37.1 | 18.2 | 28.3 |
| $k = 50$ | 32.8 | 25.0 | 37.0 | 18.1 | 28.2 |

Table 5: Combination of three strategies.

| | Module | | | mIoU | $\Delta$ |
|---|---|---|---|---|---|
| | *ATR* | *SSR* | *SHE* | | |
| baseline | – | – | – | 27.5 | – |
| | ✓ | ✓ | – | 28.6 | +1.1 |
| | – | ✓ | ✓ | 28.9 | +1.4 |
| | ✓ | – | ✓ | 28.9 | +1.4 |
| | ✓ | ✓ | ✓ | **29.2** | **+1.7** |

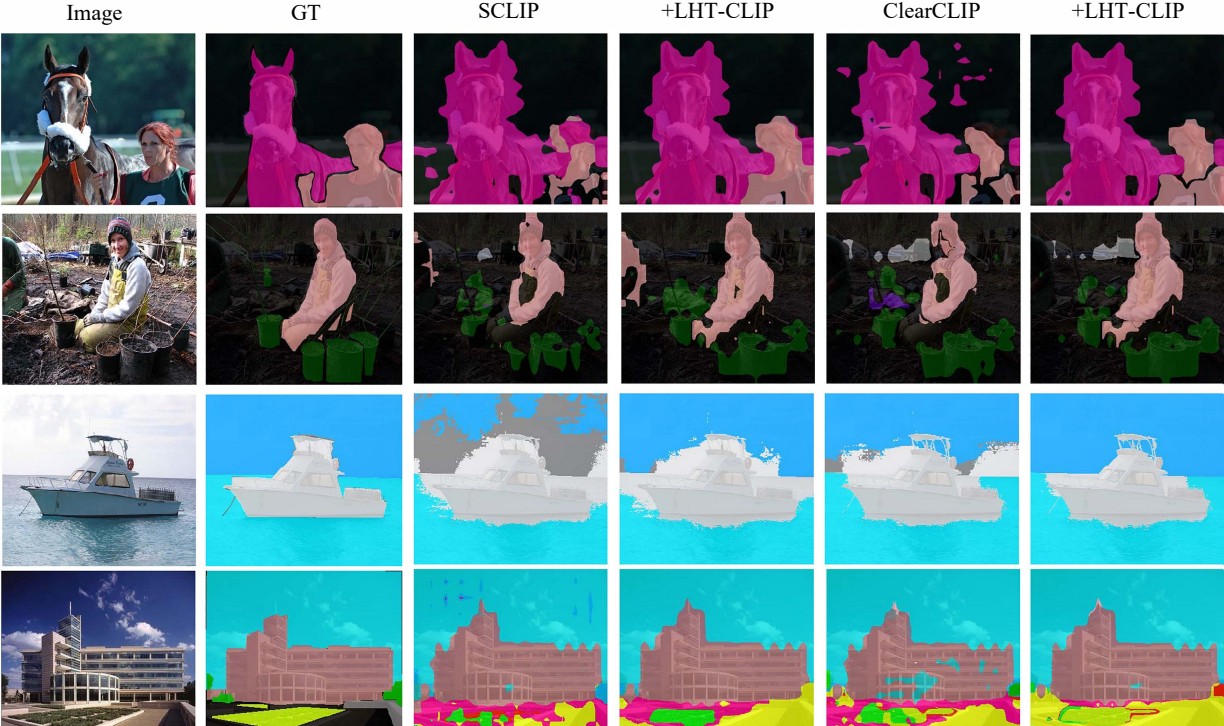

Figure 7: Qualitative comparison between CLIP-based training-free baseline methods and their counterparts integrated with LHT-CLIP.

**Study of combination effect.** In prior parts, we show the effectiveness of each individual strategy in Table 2, Table 3 and Table 4. The Table 5 further presents results of different combinations under the same settings. The result highlights the complementary contributions of each strategy to the overall segmentation performance, yielding average 1.7 mIoU improvement on these four datasets.

### 4.3 Qualitative results

In Figure 7, we present a qualitative comparison between CLIP-based training-free baseline methods and their counterparts integrated with LHT-CLIP. As shown in the figure, the integration of LHT-CLIP consistently leads to more accurate and visually coherent segmentation results. The enhanced models produce cleaner segmentation maps with reduced noise, improved spatial consistency among same-category objects, and better delineation of object boundaries, all while preserving the original text-image alignment capabilities of the baseline methods.

### 4.4 Applicability beyond CLIP

Table 6: Performance of the ViT-B variant as the SigLIP vision encoder. The best results are obtained when combined with our proposed LHT-CLIP method, as highlighted in `gray` .

| Method | VOC20 | City | C59 | ADE | Stuff | Avg |
|---|---|---|---|---|---|---|
| SigLIP | 48.0 | 20.5 | 18.6 | 11.5 | 12.1 | 22.1 |
| SigLIP+ClearCLIP | 5.7 | 2.9 | 1.7 | 0.6 | 1.6 | 2.3 |
| SigLIP+LHT-CLIP | 59.1 | 23.2 | 23.1 | 13.8 | 15.3 | 26.9 |

Table 7: Efficiency comparison of individual strategies.

| Models | FLOPs(G) ↓ | Params(M) ↓ | Speed(FPS) ↑ |
|---|---|---|---|
| CLIP | 106.10 | 149.6 | 13.7 |
| ResCLIP | 141.34 | 149.6 | 3.0 |
| baseline | 100.70 | 149.6 | 13.9 |
| +ATR | 100.88 | 149.6 | 12.9 |
| +SSR | 100.88 | 149.6 | 11.7 |
| +SHE | 102.65 | 149.6 | 8.2 |

To evaluate the generalizability of our approach beyond CLIP, we further assess it on the SigLIP model (Zhai et al., 2023). We adopt the ViT-B variant as the SigLIP vision encoder and conduct experiments on five datasets without a background category, as summarized in Table 6. Similar to CLIP, the vanilla SigLIP model performs poorly, achieving an average score of only 22.1. Notably, prior training-free methods such as ClearCLIP (Lan et al., 2024a) perform even worse than vanilla SigLIP. We attribute this to architectural differences between CLIP and SigLIP, in particular the use of an AttentionPool projector in SigLIP instead of linear projection. In this setting, modifying the final transformer layer can significantly disrupt the input distribution to the projector and degrade performance. We therefore adopt vanilla SigLIP as the baseline.

In contrast to prior approaches that focus exclusively on modifying the final transformer layer, our method is guided by a fine-grained analysis of visual discriminability and semantic alignment across token, head, and layer levels. Rather than restricting improvements to the last stage, we introduce targeted modifications at earlier layers of the model, leading to more robust and generalizable inference. Despite the architectural and training-objective differences, LHT-CLIP consistently improves open-vocabulary segmentation performance on SigLIP without modifying the model or introducing additional training. As shown in Table 6, our approach delivers a notable gain of 4.8 points in average segmentation performance, improving from 22.1 to 26.9. This result demonstrates that our LTH-CLIP framework is not unique to CLIP, but reflects more general properties of ViT-based vision–language models and captures model-agnostic structural phenomena rather than CLIP-specific artifacts. Importantly, the same hyperparameter settings (e.g., sparsity threshold $\tau$, SSR layer range and coefficient $(l_{start}, l_{end}, \alpha)$, number of selected heads $k$, and similarity threshold $\beta$) are reused when transferring from CLIP to SigLIP, without task- or model-specific tuning.

## 4.5 Efficiency Comparison

In Table 7, we report efficiency results on the Context59 dataset using an A5000 GPU with the ViT-B/16 backbone, taking ClearCLIP as the baseline. The results demonstrate that the inference-time overhead introduced by LHT-CLIP is modest. As shown in Table 11, integrating all three components (ATR, SSR, and SHE) leads to only a minor increase in inference latency compared to standard CLIP-based baselines, while remaining substantially more efficient than the prior state-of-the-art ResCLIP. Notably, by removing the computationally expensive PAMR post-processing step used in ResCLIP's Semantic Feedback Refinement module, LHT-CLIP simultaneously improves segmentation performance and reduces computational cost. As a result, the inference speed increases from 3.0 FPS to 8.2 FPS, and the total computational cost is reduced from 141.34G to 102.65G FLOPs.

# 5 Conclusion

In this paper, we present a comprehensive analysis of the visual discriminability of pretrained CLIP models at the token, layer, and head levels. Based on the analysis, we introduce LHT-CLIP, a training-free framework that enhances visual discriminability while preserving semantic alignment. LHT-CLIP comprises three complementary components: (1) abnormal token replacement, (2) spatial–semantic reweighting, and (3) selective head enhancement. Our approach provides lightweight, plug-and-play modules compatible with existing architectures. Extensive experiments on multiple segmentation benchmarks show the effectiveness of LHT-CLIP outperforms strong baselines. Furthermore, since CLIP vision encoders are often frozen in MLLM training, our findings offer practical guidance for improving visual understanding in broader MLLMs.

## Acknowledgment

The authors would like to acknowledge support from NSF grants IIS-2312840 and IIS-2402952.

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

# A Appendix

In the appendix, we first provide additional related literature in Section A.1. We then present extended experimental details and results for both the ViT-B/16 and ViT-L/14 vision encoders in Section A.2. For each model, we report the distribution of average head visual discriminability scores across multiple datasets and conduct a component-wise analysis of their contributions to visual discriminability. In Section A.3, we address key concerns, including hyperparameter selection, applicability beyond CLIP, and a comparison between SSR and direct skip methods.

## A.1 Additional literature

**Vision-language pre-training models.** Deep learning is experiencing a significant paradigm shift driven with the emergence of large-scale vision-language models. Among these, the Contrastive Language-Image Pre-Training (CLIP) framework (Radford et al., 2021) has achieved remarkable success, largely attributed to its strong zero-shot and few-shot generalization capabilities in visual recognition tasks, particularly image classification. Building on these strengths, an expanding body of research has sought to further improve CLIP's zero-shot performance through enhanced training on large-scale image-text pairs (Jia et al., 2021; Cherti et al., 2023; Schuhmann et al., 2022; Zhai et al., 2023), or by enabling its efficient adaptation to novel downstream tasks using limited labeled data (Gao et al., 2024; Silva-Rodriguez et al., 2024; Sung et al., 2022; Yu et al., 2023; Zhang et al., 2021). Nevertheless, as CLIP is pre-trained predominantly at the image level, its learned representations, particularly the [CLS] token, are optimized to capture global semantics. This coarse-grained supervision inherently limits their utility in dense prediction tasks, where fine-grained spatial localization is critical.

**Open-vocabulary Semantic Segmentation.** Driven by the remarkable generalization capabilities of large-scale vision-language models (Radford et al., 2021; Jia et al., 2021; Cherti et al., 2023; Schuhmann et al., 2022), a growing line of research has focused on open-vocabulary semantic segmentation (Ren et al., 2023; Liang et al., 2023; Luo et al., 2023a;b; Shin et al., 2022; Xu et al., 2023b;a; Li et al., 2024a; Qorbani et al., 2025), which seeks to extend global cross-modal alignment to fine-grained, pixel-level predictions. Existing approaches can be broadly categorized into three groups: *fully supervised*, *weakly supervised*, and *training-free* methods, based on the level of auxiliary supervision required for adaptation. *Fully-supervised* approaches (Liang et al., 2023; Luo et al., 2023b; Li et al., 2024a; Qorbani et al., 2025) adapt pretrained CLIP models to semantic segmentation by leveraging large-scale datasets with dense pixel-wise annotations for a predefined, yet limited, set of categories. Although such supervision facilitates the learning of fine-grained spatial representations, these methods often struggle to generalize to novel categories. Furthermore, their reliance on labor-intensive, densely annotated datasets poses significant challenges to scalability, limiting their applicability in real-world scenarios. Instead of requiring pixel-wise annotations, *weakly-supervised* approaches leverage auxiliary datasets with image-level annotations to adapt pre-trained CLIP models for semantic segmentation. These approaches commonly employ large-scale image-text corpora, wherein textual descriptions explicitly reference the object categories present in the corresponding images. Adaptation is achieved by enforcing cross-modal alignment through a contrastive loss (Xu et al., 2022a), analogous to the original CLIP pre-training objective. Although such strategies alleviate the burden of dense supervision, they still rely on access to large-scale annotated datasets. However, the auxiliary datasets are substantially smaller than those employed during CLIP pre-training, which limits the generalization capability. Additionally, these methods presuppose prior knowledge of the categories present in each image, thereby constraining their applicability in genuinely open-world segmentation scenarios.

**Training-free open-vocabulary semantic segmentation.** *Training-free* methods explore the feasibility of employing frozen CLIP models to generate segmentation masks in the absence of additional data for adaptation. Some approaches rely on auxiliary models pre-trained on large-scale datasets such as DINO (Wysoczańska et al., 2024; Lan et al., 2024b), SAM (Zhang et al., 2025), or Stable Diffusion (Corradini et al., 2024; Sun et al., 2024), which incur significant computational and memory overhead. An alternative line of research seeks to enhance CLIP's capability for dense visual representation by modifying the inference pipeline of its visual encoder. For instance, MaskCLIP (Zhou et al., 2022a) removes the self-attention module in the

final Transformer layer and demonstrates that the resulting value embeddings encode local visual features that align effectively with textual prompts. CLIP-Surgery (Li et al., 2023) introduces a dual-path structure that replaces the original self-attention with value-value attention to better preserve semantic consistency, mitigating the tendency to attend to unrelated regions. Building on this idea, GEM (Bousselham et al., 2024) and SCLIP (Wang et al., 2024) generalize the approach by incorporating correlative self-attention mechanisms, such as query-query and key-key interactions. ClearCLIP (Lan et al., 2024a) further simplifies the architecture by removing the final feed-forward layer and associated residual connections, which were found to contribute to noisy segmentation outputs. Additionally, NACLIP (Hajimiri et al., 2025) imposes explicit spatial regularization, encouraging each token to primarily attend to its neighbors to improve the spatial coherence.

Taking into account both generalization ability and computational costs, this work aims to advance existing training-free methods for open-vocabulary semantic segmentation. While prior training-free approaches predominantly rely on features extracted from the final layer of the visual encoder, we instead conduct a comprehensive analysis across token, head, and layer levels. Building on this multi-level analysis, we propose recipe at each level to enhance the spatial representation capacity of CLIP, thereby improving segmentation performance within a training-free framework.

## A.2 Additional experimental results

### A.2.1 Additional Implementation Details.

We adopt CLIP (Radford et al., 2021) with ViT-B/16 and ViT-L/14 backbones, implemented in MMSegmentation (Contributors, 2020). Following the protocol of Yang et al. (2024), input images are resized to 336 pixels on the shorter side, except for Cityscapes, which is resized to 560 pixels to accommodate higher resolution. Inference is conducted with a sliding-window strategy using $224 \times 224$ crops and a stride of 112 pixels. Consistent with TCL (Cha et al., 2023), we avoid computationally intensive post-processing methods that hinder fair comparison, such as PAMR (Araslanov & Roth, 2020) (used in TCL (Cha et al., 2023), NACLIP (Hajimiri et al., 2025)) and DenseCRF (Krähenbühl & Koltun, 2011) (used in ReCo (Shin et al., 2022)). For textual inputs, we employ the standard ImageNet prompts (Radford et al., 2021) without additional prompting strategies. Based on the ablation study in Section 4 and Section A.2.4, the following configuration is fixed across all dataset evaluations without task-specific tuning

- **ViT-B/16**: The hoyer threshold is set to $\tau = 0.5$, and the reweighting operation is applied to layers $[10, 11]$ (i.e., the two layers preceding the final one), with a reweighting coefficient of 0.1. The top-10 attention heads with the highest visual discriminability are selected using a filter threshold of $\beta = 0.7$, corresponding to the following (layer, head) pairs: (8,9), (8,8), (7,10), (9,12), (7,3), (9,4), (5,1), (9,6), (4,11), and (8,6).
- **ViT-L/14**: The hoyer threshold is set to $\tau = 0.4$ and the reweighting is applied to layers $[17, 23]$ with a coefficient of 0.1. Top-performing heads are selected using the same threshold $\beta = 0.7$, with the top-30 (layer, head) pairs identified as: (11, 3), (9, 3), (7, 9), (11, 6), (10, 10), (9, 13), (3, 10), (4, 14), (10, 6), (6, 9), (7, 12), (14, 16), (11, 8), (10, 13), (8, 4), (8, 8), (10, 8), (9, 4), (2, 11), (9, 6), (8, 1), (14, 1), (16, 2), (4, 13), (13, 11), (11, 14), (7, 4), (14, 11), (13, 13), and (3, 13).

All experiments are conducted using eight NVIDIA RTX A5000 GPUs, each with 24 GB of memory.

### A.2.2 Effects of individual components effects on visual discriminability

**Complementary and effective for visual discriminability.** The ablation study in Section 4 confirms the effectiveness of each individual strategy for segmentation performance. As shown in Figure 8, we further analyze their contributions to visual discriminability by examining how progressively integrating these strategies improves the quality of the final representations. Similar to the definition of visual discriminability, we treat the token-wise similarity score as the output of a binary classifier predicting whether two tokens belong to the same category, and plot the ROC curves of penultimate-layer features. A higher area under the curve (AUC) indicates stronger visual discriminability. The results show that each strategy individually improves AUC, while their combination yields substantial gains in visual discriminability. For the ViT-B/16 backbone,

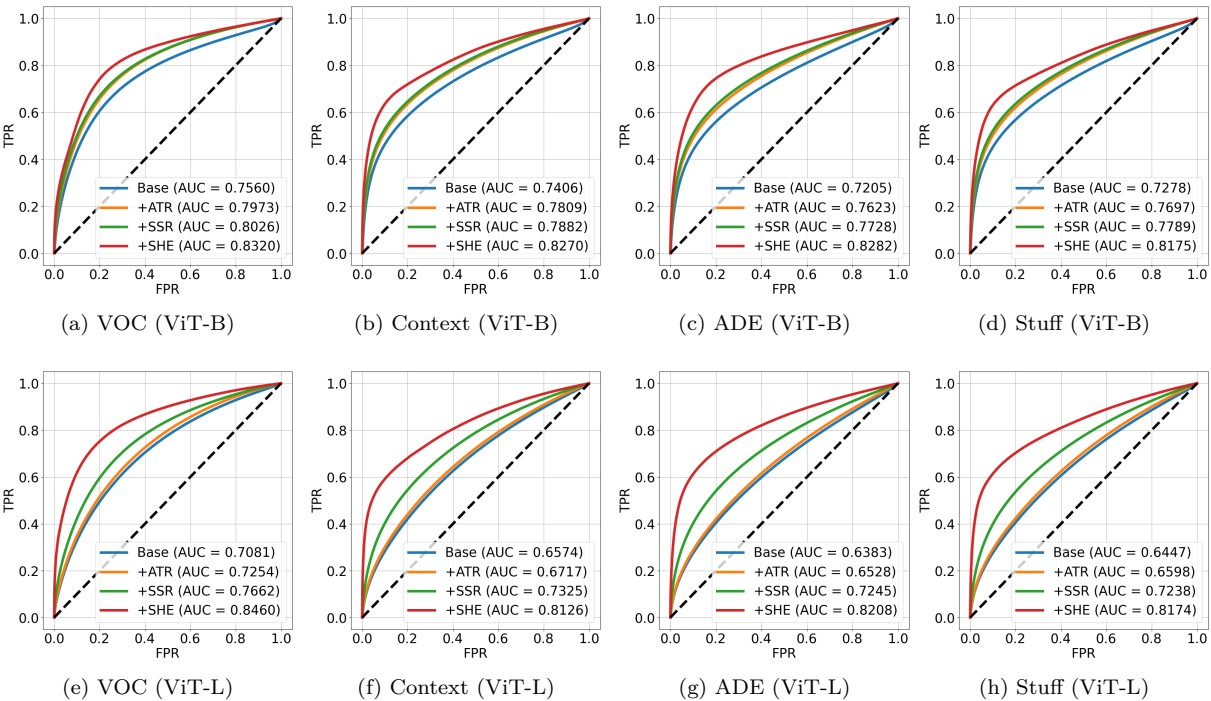

Figure 8: ROC curves of the penultimate-layer output features for ViT-B/16 and ViT-L/14 backbones across four datasets. Each curve corresponds to an incremental combination of strategies. The results demonstrate consistent improvements in AUC with each added strategy, culminating in significantly higher visual discriminability when all are applied.

AUC increases from 0.7560 to 0.8320 on VOC, 0.7406 to 0.8270 on Context, 0.7205 to 0.8282 on ADE, and 0.7278 to 0.8175 on COCO-Stuff. Similarly, for ViT-L/14, AUC rises from 0.7081 to 0.8460 on VOC, 0.6574 to 0.8126 on Context, 0.6383 to 0.8208 on ADE, and 0.6447 to 0.8174 on COCO-Stuff. These improvements highlight the complementary nature of the proposed strategies and their collective effectiveness in enhancing visual discriminability across diverse benchmarks.

### A.2.3    Head AUC distributions

In Section 3, we presented the distribution of a subset of head-level visual discriminability scores across multiple datasets, employing ViT-B/16 as the visual encoder. In practice, we observed that the discriminability of head features is also degraded by the presence of abnormal tokens in the final layers. To fully realize the potential of head features, therefore, we apply abnormal token replacement to head features. Similarly, we consistently find that certain heads exhibit significantly higher visual discriminability than others. Since we select the top-$k$ heads based on their average visual discriminability across multiple datasets, we additionally present the distribution of average visual discriminability scores for all heads in descending order in Figure 9 for ViT-B/16 and Figure 10 for ViT-L/14, to facilitate reference and reuse in future research.

### A.2.4    Additional experiments on ViT-L/14

In Table 8, we present a detailed comparison of open-vocabulary semantic segmentation models on multiple benchmarks using the ViT-L/14 backbone. LHT-CLIP consistently improves the performance of leading methods, including ClearCLIP (Lan et al., 2024a), NACLIP (Hajimiri et al., 2025), and ResCLIP (Yang et al., 2024). In particular, integrating LHT-CLIP with ResCLIP achieves state-of-the-art results. As a plug-and-play module, LHT-CLIP provides consistent gains across all datasets, highlighting its robustness and generalization. As noted in Yang et al. (2024), many methods suffer performance drops exceeding 2% mIoU when switching from ViT-B/16 to ViT-L/14; for example, ClearCLIP declines by 2.7%. In contrast, LHT-

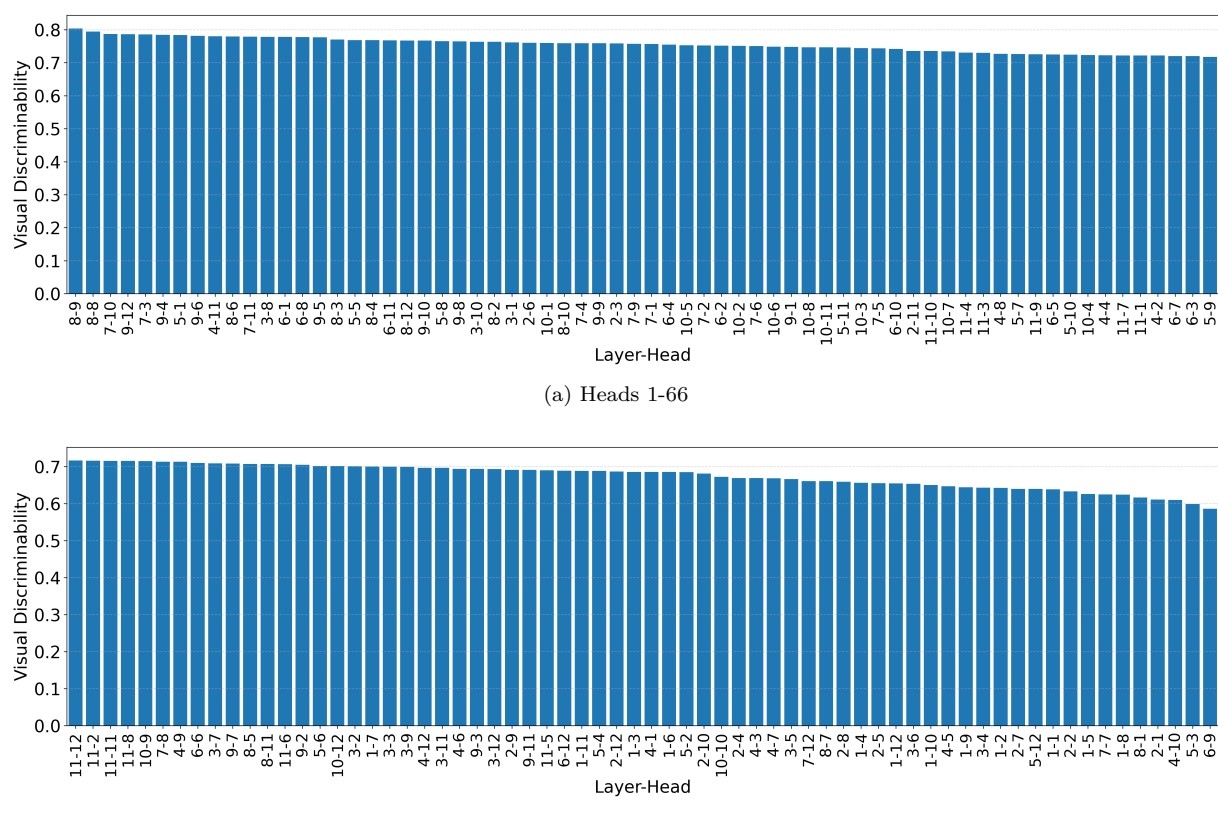

(a) Heads 1-66

(b) Heads 67-172

Figure 9: Distribution of average visual discriminability score for individual attention heads across three datasets (Context, ADE, and Stuff) using the ViT-B/16 model, with the heads from final layer excluded. Bars are sorted in descending order of average visual discriminability.

CLIP largely mitigates this degradation, demonstrating strong cross-backbone robustness. These results further validate its effectiveness in enhancing open-vocabulary segmentation across both ViT-B/16 and ViT-L/14 architectures.

Additionally, in Table 9, Table 10, Table 11, and Table 12, we present detailed analyses of the hoyer threshold parameter, the spatial-semantic reweighting configuration, the number of selected attention heads, and the ablation analysis of individual components, respectively, for the ViT-L/14 model. Consistent with the ViT-B/16 results, the optimal configuration is achieved with a moderate sparsity level ($\tau = 0.4$), a reweighting coefficient $\alpha = 0.1$ applied to layers 17–23, and the top-$k$ heads with $k = 30$. Therefore, we adopt these settings as fixed hyperparameters for ViT-L/14 across all benchmarks, without any dataset-specific tuning. Moreover, the ablation study confirms the effectiveness of each individual component and their complementary effect. As shown in Table 12, their combination leads to a substantial improvement of 5.1 mIoU, achieving a final score of 30.1 mIoU across the four benchmark datasets.

### A.3 Further Discussion

### A.3.1 Hyperparameter Selection

While LHT-CLIP introduces several hyperparameters, these are guided by universal patterns consistently observed across datasets, such as the sparsity of abnormal tokens, the trade-off between visual discriminability and semantic alignment, and the presence of shared visually discriminative heads. Building on these consistent phenomena, we emphasize three key points:

Table 8: Performance comparison of our approach with other methods on eight semantic segmentation benchmarks following the evaluation protocol in Section 4. Our results are marked in gray.

| Methods | Training | With a background class | | | Without background class | | | | | Avg. |
|---|---|---|---|---|---|---|---|---|---|---|
| | | VOC21 | C60 | Object | VOC20 | City | C59 | ADE | Stuff | |
| CLIP | ✗ | 9.8 | 4.2 | 3.9 | 18.8 | 4.0 | 5.6 | 2.1 | 2.8 | 6.4 |
| MaskCLIP | ✗ | 24.4 | 10.0 | 9.9 | 30.0 | 11.8 | 12.6 | 7.8 | 10.1 | 14.6 |
| CLIPSurgery | ✗ | 47.7 | 27.2 | 27.5 | 80.5 | 32.1 | 31.5 | 16.4 | 17.3 | 35.0 |
| SCLIP | ✗ | 44.4 | 22.3 | 24.9 | 60.3 | 32.2 | 20.5 | 7.1 | 13.1 | 28.1 |
| ClearCLIP | ✗ | 48.7 | 28.3 | 29.7 | 80.0 | 27.9 | 29.6 | 15.0 | 19.9 | 34.9 |
| +LHT-CLIP (ours) | ✗ | 62.8 | 33.9 | 33.5 | 85.9 | 40.5 | 39.0 | 20.3 | 26.1 | 42.7 (+7.8) |
| NACLIP (Hajimiri et al., 2025) | ✗ | 52.2 | 28.7 | 30.4 | 78.7 | 31.4 | 32.1 | 17.3 | 21.4 | 36.5 |
| +LHT-CLIP (ours) | ✗ | 63.4 | 34.0 | 33.4 | 84.6 | 40.8 | 39.0 | 20.5 | 25.9 | 42.7 (+6.2) |
| ResCLIP (Yang et al., 2024) | ✗ | 55.7 | 29.7 | 30.8 | 81.3 | 32.8 | 33.6 | 17.9 | 22.9 | 38.1 |
| +LHT-CLIP (ours) | ✗ | 63.4 | 34.0 | 33.4 | 85.9 | 40.9 | 39.0 | 20.5 | 26.0 | 42.9 (+4.8) |

Table 9: Study of hoyer sparsity threshold $\tau$.

| $\tau$ | C60 | Stuff | C59 | ADE | Avg |
|---|---|---|---|---|---|
| $\tau = 0.2$ | 2.4 | 2.8 | 3.8 | 2.3 | 2.8 |
| $\tau = 0.4$ | 29.7 | 22.5 | 33.6 | 17.8 | 25.9 |
| $\tau = 0.5$ | 29.2 | 22.2 | 33.3 | 17.6 | 25.6 |
| $\tau = 0.8$ | 28.9 | 22.0 | 33.0 | 17.4 | 25.3 |
| $\tau = 0.9$ | 28.9 | 22.0 | 33.0 | 17.4 | 25.3 |
| baseline | 28.6 | 21.7 | 32.5 | 17.3 | 25.0 |

Table 10: Study of $(l_{\mathrm{start}}, l_{\mathrm{end}}, \alpha)$ in SSR module.

| $(l_{\mathrm{start}}, l_{\mathrm{end}}, \alpha)$ | C60 | Stuff | C59 | ADE | Avg |
|---|---|---|---|---|---|
| baseline | 28.6 | 21.7 | 32.5 | 17.3 | 25.0 |
| (14, 23, 0.1) | 30.8 | 21.2 | 33.9 | 16.9 | 25.7 |
| (17, 23, 0.1) | 31.4 | 23.1 | 34.7 | 17.8 | 26.8 |
| (19, 23, 0.1) | 30.9 | 23.0 | 34.2 | 17.9 | 26.5 |
| (17, 23, 0.05) | 30.3 | 22.5 | 33.6 | 17.5 | 26.0 |
| (17, 23, 0.2) | 31.6 | 22.1 | 34.8 | 17.8 | 26.6 |

1. **Generalizable.** The hyperparameters are generalizable across datasets. In practice, we found that values selected on a small subset of training data (e.g., 1,000 randomly selected training samples from {Context, ADE, COCO-Stuff}) also perform well on other datasets, indicating that they are not overfitted to specific datasets. As shown in Section 4.1 and Section A.2.4, we apply the same hyperparameters to all datasets without specific tuning, underscoring their strong generalizability.

2. **Robustness.** The method is robust to a wide range of hyperparameter values, which means that precise tuning is not required to achieve strong performance. This is demonstrated in our ablation studies Section 4.2 and Section A.2.4, where the results remain consistent in various hyperparameter settings.

3. **Reliable initialization.** Additionally, our analysis provides interpretable heuristics for selecting hyperparameters. For example, in the case of an SSR starting layer, one can use the observed turning point in visual discriminability or semantic alignment as a reliable initialization.

Taken together, while several hyperparameters are introduced, we believe these properties support the practical usability of our method without requiring extensive or fragile hyperparameter tuning.

### A.3.2 Comparison between SSR and direct skip methods

Recent work on the Perception Encoder (Bolya et al., 2025) highlights the effectiveness of directly selecting an earlier layer's output for segmentation. This approach is conceptually related to our spatial–semantic reweighting (SSR) strategy, which emphasizes residual pathways by amplifying their contribution to counteract the dominance of deeper layers. In Table 13, we present a systematic comparison between our SSR method and the direct-skip strategy, using ViT-L/14 as the CLIP vision encoder and adopting ClearCLIP as the baseline.

Although directly leveraging earlier layer outputs provides some improvement over baseline CLIP performance, our experiments show that SSR consistently achieves superior results. For example, when augmented with ATR and SHE, directly skipping layers 20–23 of the ViT-L encoder and feeding the output of layer 19 into layer 24 yields an average performance of 40.8, higher than the baseline ClearCLIP score of 34.9, but still below the performance of our full LHT-CLIP method (42.7), as reported in Table 13.

Table 11: Study of number of selected heads $k$.

| $k$ | C60 | Stuff | C59 | ADE | Avg |
|---|---|---|---|---|---|
| baseline | 28.6 | 21.7 | 32.5 | 17.3 | 25.0 |
| $k = 1$ | 32.0 | 24.3 | 35.0 | 18.6 | 27.5 |
| $k = 10$ | 32.8 | 24.6 | 36.0 | 19.2 | 28.2 |
| $k = 30$ | 33.1 | 24.9 | 36.3 | 19.5 | 28.5 |
| $k = 60$ | 33.1 | 24.9 | 36.3 | 19.5 | 28.5 |
| $k = 100$ | 33.0 | 24.8 | 36.4 | 19.5 | 28.4 |

Table 12: Combination of three strategies.

| Methods | Module | | | mIoU | $\Delta$ |
|---|---|---|---|---|---|
| | ATR | SSR | SHE | | |
| baseline | – | – | – | 25.0 | – |
| | ✓ | ✓ | – | 28.1 | +3.1 |
| | – | ✓ | ✓ | 29.2 | +4.2 |
| | ✓ | – | ✓ | 29.0 | +4.0 |
| Ours | ✓ | ✓ | ✓ | **30.1** | **+5.1** |

Table 13: Comparison between SSR and direct skip methods using the ClearCLIP as the baseline method and ViT-L/14 as the CLIP vision encoder. The best results are obtained when combined with our proposed LHT-CLIP method, as highlighted in gray .

| Method | VOC21 | C60 | Obj | VOC20 | City | C59 | ADE | Stuff | Avg |
|---|---|---|---|---|---|---|---|---|---|
| ClearCLIP | 48.7 | 28.3 | 29.7 | 80.0 | 27.9 | 29.6 | 15.0 | 19.9 | 34.9 |
| ClearCLIP+direct-skip | 60.3 | 33.7 | 32.1 | 77.3 | 40.8 | 38.8 | 20.1 | 23.7 | 40.8 |
| ClearCLIP+LHT-CLIP | 62.8 | 33.9 | 33.5 | 85.9 | 40.5 | 39.0 | 20.3 | 26.1 | 42.7 |

### A.3.3 Comparison of Norm-Based and Sparsity-Based Criteria for ATR

We regard ATR as a flexible framework that functions by detecting and replacing emergent abnormal tokens. As discussed in Section 2, these tokens exhibit two distinctive characteristics: unusually large activation magnitudes and sparse activation patterns. To evaluate the effectiveness of a magnitude-based criterion for abnormal token detection, we conduct additional experiments using ViT-B/16 as the vision encoder, with ClearCLIP as the baseline method.

The norm-based approach identifies abnormal patches by computing the activation norm of each visual patch, classifying those above a threshold $\gamma$ as abnormal. Empirically, we find that this method achieves its best average performance of 27.9 across the four datasets used in Table 14 when the threshold is set to $\gamma = 14$, which is comparable to the performance of our sparsity-based ATR method (28.0). However, the norm-based method is more sensitive to threshold selection. For example, the overall performance drops to 26.8 when $\gamma = 12$ and 27.6 when $\gamma = 16$, whereas the sparsity-based approach remains more stable under hyperparameter variations in Table 2.

### A.3.4 Extension to Monocular Depth Estimation

While this work primarily focuses on training-free open-vocabulary semantic segmentation, we further investigate the generality of our findings by extending LHT-CLIP to monocular depth estimation. In particular, we conduct additional experiments based on DepthCLIP (Zhang et al., 2022), a training-free framework that adapts CLIP representations for depth prediction. This extension allows us to examine whether our analysis of visual discriminability and the proposed components remain effective beyond semantic segmentation.

**Local and Global Roles of LHT-CLIP Components.** From a methodological perspective, the three components of LHT-CLIP can be categorized according to their operational scope:

Table 14: Study of norm threshold $\gamma$. The best results and corresponding $\gamma$ are highlighted in gray .

| $\tau$ | C60 | Stuff | C59 | ADE | Avg |
|---|---|---|---|---|---|
| $\gamma = 12$ | 31.7 | 23.0 | 35.7 | 16.9 | 26.8 |
| $\gamma = 13$ | 32.7 | 23.9 | 36.6 | 17.5 | 27.7 |
| $\gamma = 14$ | 32.8 | 24.2 | 36.8 | 17.8 | 27.9 |
| $\gamma = 15$ | 32.6 | 24.3 | 36.7 | 17.8 | 27.8 |
| $\gamma = 16$ | 32.2 | 24.1 | 36.6 | 17.6 | 27.6 |
| baseline | 32.4 | 24.0 | 36.0 | 17.6 | 27.5 |

Table 15: Effect of individual LHT-CLIP components on monocular depth estimation performance. Red indicates improvement over the baseline, and blue indicates degradation.

| Method | $\delta < 1.25 \uparrow$ | $\delta < 1.25^2 \uparrow$ | $\delta < 1.25^3 \uparrow$ | rel$\downarrow$ | $\log_{10} \downarrow$ | rmse$\downarrow$ |
|---|---|---|---|---|---|---|
| DepthCLIP | 0.373 | 0.672 | 0.850 | 0.367 | 0.160 | 1.191 |
| DepthCLIP+ATR | 0.378 | 0.675 | 0.851 | 0.375 | 0.159 | 1.182 |
| DepthCLIP+SSR | 0.346 | 0.640 | 0.833 | 0.353 | 0.169 | 1.246 |
| DepthCLIP+SHE($\beta = 0.9$) | 0.383 | 0.675 | 0.850 | 0.405 | 0.159 | 1.171 |
| DepthCLIP+SHE($\beta = 0.5$) | 0.383 | 0.664 | 0.838 | 0.445 | 0.161 | 1.175 |
| DepthCLIP+ATR+SHE($\beta = 0.9$) | 0.384 | 0.675 | 0.851 | 0.404 | 0.159 | 1.170 |

(1). Abnormal Token Replacement (ATR) acts as a **local** mechanism by replacing anomalous tokens through neighborhood aggregation, thereby enhancing spatial coherence.

(2). Spatial-Semantic Reweighting (SSR) functions as a **global** mechanism that improves visual discriminability by rebalancing semantic and visual signals across layers.

(3). Selective Head Enhancement (SHE) can operate in **either a local or global** manner depending on the similarity threshold. A larger threshold emphasizes localized, structure-preserving consistency, while a smaller threshold enforces broader semantic coherence across spatially distant regions.

This categorization provides a useful lens for understanding how different dense prediction tasks benefit from local versus global mechanisms.

**Task-Specific Effects on Depth Estimation.** To evaluate our method on monocular depth estimation, we fellow the default settings of DepthCLIP (Zhang et al., 2022) on NYU Depth v2 dataset (Silberman et al., 2012) and adopt ViT/B-16 model as the vision encoder. We follow the same evaluation protocol of DepthCLIP (Zhang et al., 2022) and assess performance using four standard metrics: mean absolute relative error (rel), root mean squared error (rmse), absolute error in log space ($\log_{10}$), and threshold accuracy ($\delta_i$). These metrics are defined as follows:

$$\text{rel} = \frac{1}{n} \sum_p \frac{|y_p - \hat{y}_p|}{y_p}, \tag{9}$$

$$\text{rmse} = \sqrt{\frac{1}{n} \sum_p (y_p - \hat{y}_p)^2}, \tag{10}$$

$$\log_{10} = \frac{1}{n} \sum_p |\log_{10}(y_p) - \log_{10}(\hat{y}_p)|, \tag{11}$$

$$\delta_i = \text{percentage of pixels } p \text{ satisfying } \max\left(\frac{y_p}{\hat{y}_p}, \frac{\hat{y}_p}{y_p}\right) < \text{thr}, \tag{12}$$

where thr $\in \{1.25, 1.25^2, 1.25^3\}$ and $n$ denotes the number of valid pixels.

As shown in Table 15, monocular depth estimation, similar to semantic segmentation, requires spatially coherent and locally discriminative visual features across both foreground and background regions. Consistent with this requirement, ATR and SHE with a large similarity threshold (e.g., 0.9), both operating primarily at a local level, lead to clear improvements in depth prediction quality across most evaluation metrics, with the exception of the mean absolute relative error (rel). These gains stem from enhanced pixel-level visual discriminability and improved spatial consistency.

However, depth estimation differs fundamentally from semantic segmentation in its reliance on global visual discriminability. While semantic segmentation benefits from globally consistent features that enable uniform

labeling of spatially separated regions belonging to the same object category, monocular depth estimation depends more heavily on local geometric continuity. Regions that share similar semantic appearance may exhibit significantly different depth values. For example, multiple desks in a classroom may belong to the same semantic category but lie at different distances from the camera. As a result, enforcing strong feature similarity across distant regions, via global mechanisms such as SSR or SHE with a smaller similarity threshold (e.g., 0.5), can be detrimental for depth estimation, as it may oversmooth depth variations that are geometrically meaningful.

**Conclusion.** These findings highlight that the effectiveness of local and global mechanisms is task-dependent. While global semantic consistency is crucial for semantic segmentation, depth estimation benefits more from localized discriminability and spatial continuity. Importantly, the observed behaviors are consistent with our core analysis of visual discriminability at the token, head, and layer levels, further validating the generality of our findings beyond a single dense prediction task. This provides additional insight into how different dense vision tasks selectively benefit from local and global representations.

### A.3.5 Discussion of Failure Case:

In Figure 7, our qualitative results highlight typical improvements in spatial coherence and boundary quality. We acknowledge that LHT-CLIP may still struggle in certain scenarios, including (i) small or fine-grained objects, where individual patch predictions are dominated by majority voting (e.g., the green car in the ground-truth segmentation map shown in the fourth row of Figure 7), and (ii) inherently high semantic ambiguity, where CLIP representations are insufficiently discriminative (e.g., the black ground region in Figure 7, which exhibits semantic ambiguity and is misclassified into other categories). Importantly, such failure modes are not unique to our method but are shared by most training-free CLIP-based segmentation approaches that rely on frozen representations.

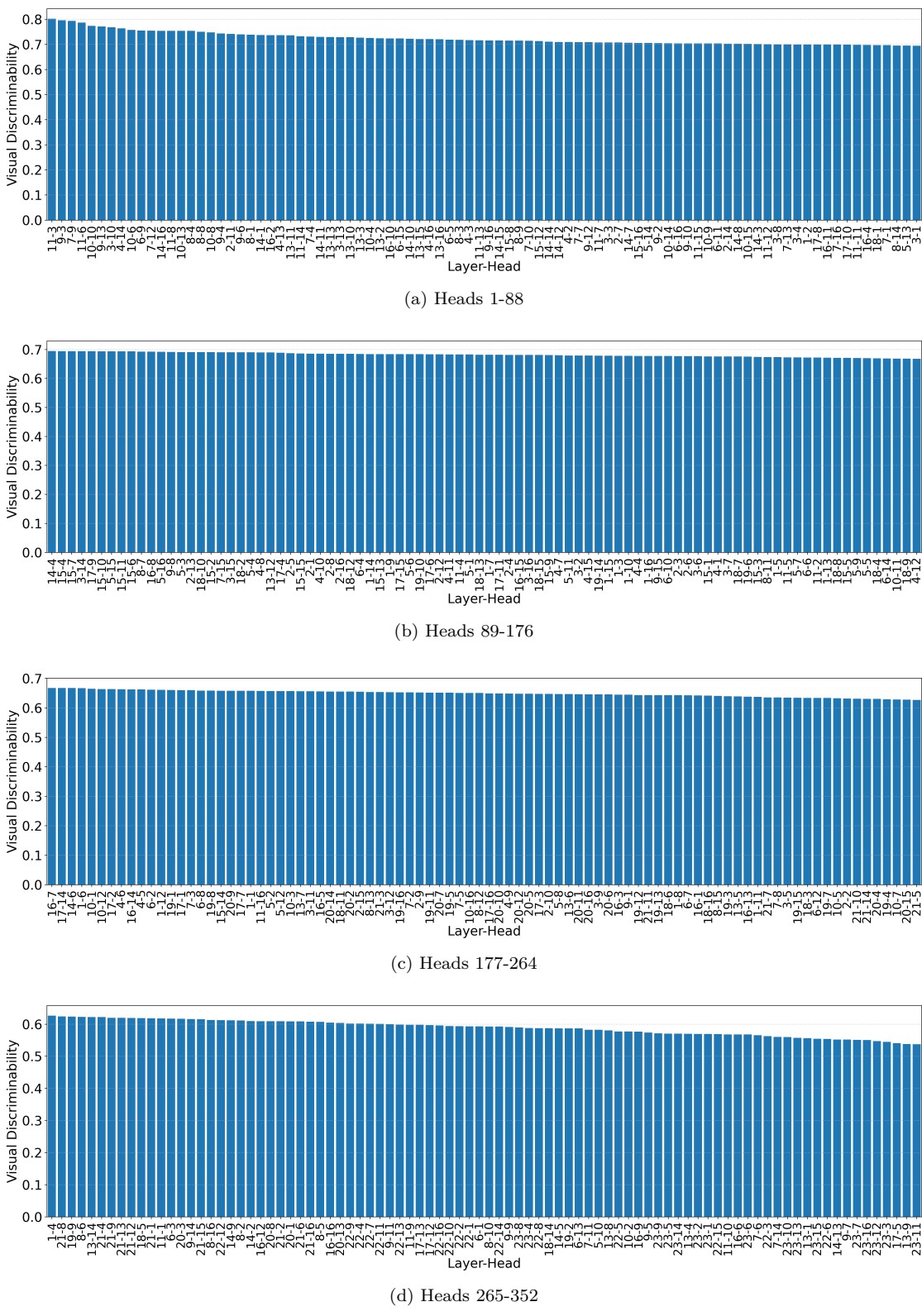

Figure 10: Distribution of average visual discriminability score for individual attention heads across three datasets (Context, ADE, and Stuff) using the ViT-L/14 model, with the heads from final layer excluded. Bars are sorted in descending order of average visual discriminability.

