# OpenReview forum: "Improving Visual Discriminability of CLIP for Training-Free Open-Vocabulary Semantic Segmentation"
_TMLR — Accepted by TMLR_

### Review · Reviewer_REZ6 · 2025-12-15

**Summary Of Contributions:**

Contributions:
This paper addresses a limitation of CLIP-based models in training-free open-vocabulary semantic segmentation, which is the mismatch between a good image–text alignment obtained during CLIP pre-training, but a poor pixel-level visual discriminability. The authors conduct an analysis of CLIP’s vision encoder at the layer, attention-head, and token levels. They found that deeper layers substantially degrade visual discriminability while providing only marginal gains in semantic alignment. A central finding is the emergence of abnormal tokens with sparse, high-norm, and bias-like activations, which dominate attention and impair spatial coherence.

Motivated by these observations, the paper proposes LHT-CLIP, a training-free framework that enhances visual discriminability while preserving semantic alignment. They also conduct extensive experiments on eight standard benchmarks, demonstrating that LHT-CLIP obtains significant improvements over several state-of-the-art training-free baselines, across different CLIP backbones, without relying on additional supervision, training, or auxiliary models.


Strength:
1. The paper provides a thorough empirical analysis of CLIP representations together with a reasonable motivation. It provides convincing evidence that degradation in visual discriminability, rather than insufficient semantic alignment, is a key bottleneck for dense prediction.
2. The proposed method has a training-free setting, avoiding additional datasets, optimization, or heavy computation. It aligns well with the motivation of open-vocabulary and general-purpose deployment.
3. The identification and mitigation of abnormal tokens via sparsity-based criteria is simple, interpretable, and seems to be more robust than prior anomaly-detection-based approaches.
4. The paper evaluates on a broad set of benchmarks and backbones, providing several new insights to the community.


Weakness:
1. The proposed method requires inference-time architectural modification. Modifying the intermediate transformer during inference may complicate integration with highly optimized or fixed inference pipelines.
2. The evaluation is limited to open-vocabulary semantic segmentation. I wonder if the proposed method can generalize well to other dense prediction tasks, like Monocular Depth Estimation [1,2,3] ? For example, in DepthCLIP [1], authors also mentioned that one weakness of CLIP-based monocular depth estimation might be a poor pixel-level visual discriminability, since features that are not important to image classification might be neglected during CLIP’s feature extraction. As a result, depth estimation for the background is not as good as for the foreground. Related experiments and discussion would be appreciated.


[1] Zhang, Renrui, et al. "Can language understand depth?." Proceedings of the 30th ACM International Conference on Multimedia. 2022.
[2] Auty, Dylan, and Krystian Mikolajczyk. "Learning to prompt clip for monocular depth estimation: Exploring the limits of human language." Proceedings of the IEEE/CVF International Conference on Computer Vision. 2023.
[3] Son, Eunjin, and Sang Jun Lee. "CaBins: CLIP-based adaptive bins for monocular depth estimation." Proceedings of the IEEE/CVF Conference on Computer Vision and Pattern Recognition. 2024.

**Audience:**

Yes

**Audience Explanation:**

People who work on dense prediction (segmentation, object detection, or depth estimation) could benefit from this paper, especially people who work on applying vision-language models to those tasks.

**Broader Impact Concerns:**

No concerns on ethical implications of this work.

**Claims And Evidence:**

Yes

**Claims Explanation:**

The authors conducted extensive experiments on eight standard benchmarks, demonstrating that LHT-CLIP obtains significant improvements over several state-of-the-art training-free baselines, across different CLIP backbones, without relying on additional supervision, training, or auxiliary models. Those experiments also back up the argument in the introduction.

**Requested Changes:**

As stated above in the weakness part, the evaluation is limited to open-vocabulary semantic segmentation. I wonder if the proposed method can generalize well to other dense prediction tasks, like Monocular Depth Estimation [1, 2, 3] ? For example, in DepthCLIP [1], the authors also mentioned that one weakness of CLIP-based monocular depth estimation might be a poor pixel-level visual discriminability, since features that are not important to image classification might be neglected during CLIP’s feature extraction. As a result, depth estimation for the background is not as good as for the foreground. Related experiments and discussion would be appreciated.

[1] Zhang, Renrui, et al. "Can language understand depth?." Proceedings of the 30th ACM International Conference on Multimedia. 2022.
[2] Auty, Dylan, and Krystian Mikolajczyk. "Learning to prompt clip for monocular depth estimation: Exploring the limits of human language." Proceedings of the IEEE/CVF International Conference on Computer Vision. 2023.
[3] Son, Eunjin, and Sang Jun Lee. "CaBins: CLIP-based adaptive bins for monocular depth estimation." Proceedings of the IEEE/CVF Conference on Computer Vision and Pattern Recognition. 2024.

---

> ### Author Response · Authors · 2026-01-16
> **Response to Reviewer REZ6**
>
> We thank the reviewer for the constructive and positive assessment of our paper, and we are glad that the reviewer finds (i) our analysis of the discriminability–alignment trade-off **convincing**, (ii) the training-free nature of LHT-CLIP **well aligned** with open-vocabulary deployment, and (iii) the sparsity-based identification of abnormal tokens **simple and robust**.
>
> **W1**: The proposed method requires inference-time architectural modification. Modifying the intermediate transformer during inference may complicate integration with highly optimized or fixed inference pipelines.
>
> **A1**: We agree that “inference-time architectural modification” can be a practical concern for certain highly optimized/fixed pipelines. We would like to clarify what is being modified and why the integration burden is limited in practice.
> 1. **No new parameters / no retraining / no auxiliary networks**. All three components (ATR/SSR/SHE) only reweight or post-process existing intermediate activations of the frozen CLIP ViT; they do not introduce trainable modules, extra supervision, or auxiliary pretrained models, which is central to our motivation and positioning as a training-free framework.
> 2. **The modifications are local and lightweight (graph-level), not a new architecture**. SSR performs a simple scalar reweighting between the residual and attention/FFN branches in a small number of late transformer layers. It does not alter tensor shapes, attention structure, or require re-exporting the model. ATR is applied only at the penultimate layer and conducts lightweight neighborhood aggregation on a small set of identified abnormal tokens; applying it earlier was empirically found to be detrimental. SHE aggregates a small top-k set of discriminative attention heads to construct a soft pseudo-mask for feature refinement, introducing no training and only negligible overhead relative to the ViT forward pass.
> 3. **Ablations indicate stable hyperparameters and minimal tuning overhead**. We show that the key threshold for ATR has a broad stable region ($\tau\approx0.5–0.8$) and that SSR also has a small, interpretable parameterization (a small $\alpha$ over a small layer range), which reduces engineering burden when integrating into different pipelines.
>
> To be continued...

---

> ### Author Response · Authors · 2026-01-16
> **Continued Response**
>
> **W2**: The evaluation is limited to open-vocabulary semantic segmentation. I wonder if the proposed method can generalize well to other dense prediction tasks, like Monocular Depth Estimation [1,2,3] ? For example, in DepthCLIP [1], authors also mentioned that one weakness of CLIP-based monocular depth estimation might be a poor pixel-level visual discriminability, since features that are not important to image classification might be neglected during CLIP’s feature extraction. As a result, depth estimation for the background is not as good as for the foreground. Related experiments and discussion would be appreciated.
>
> **A2**: We thank the reviewer for this insightful comment, which meaningfully strengthens the presentation and scope of our work. While the current submission focuses on training-free open-vocabulary semantic segmentation, we follow the reviewer’s suggestion and, during the rebuttal period, conduct additional experiments on monocular depth estimation based on DepthCLIP [1].
>
> | Method | δ < 1.25 ↑ | δ < 1.25² ↑ | δ < 1.25³ ↑ | rel ↓ | log₁₀ ↓ | rmse ↓ |
> |------|-------------|---------------|---------------|---------|-----------|----------|
> | DepthCLIP | 0.373 | 0.672 | 0.850 | 0.367 | 0.160 | 1.191 |
> | DepthCLIP + ATR | 0.378 | 0.675 | 0.851 | 0.375 | 0.159 | 1.182 |
> | DepthCLIP + SSR | 0.346 | 0.640 | 0.833 | 0.353 | 0.169 | 1.246 |
> | DepthCLIP + SHE (β = 0.9) | 0.383 | 0.675 | 0.850 | 0.405 | 0.159 | 1.171 |
> | DepthCLIP + SHE (β = 0.5) | 0.383 | 0.664 | 0.838 | 0.445 | 0.161 | 1.175 |
> | DepthCLIP + ATR + SHE (β = 0.9) | 0.384 | 0.675 | 0.851 | 0.404 | 0.159 | 1.170 |
>
> From a methodological perspective, we **categorize our three components into local and global mechanisms**: **ATR** (abnormal token replacement via neighborhood aggregation) acts **locally** to enhance spatial coherence; **SSR** operates **globally** to improve visual discriminability by rebalancing semantic and visual signals across layers; and **SHE** can function in **either a local or global manner** depending on the threshold, where a larger threshold emphasizes more localized, structure-preserving consistency.Based on this categorization, we observe the following:
>
> **Monocular depth estimation**, similar to semantic segmentation, requires **spatially coherent and locally discriminative** visual features across both foreground and background regions. Consistent with this requirement, ATR and SHE with a large similarity threshold (e.g., 0.9), both operating primarily at a local level, lead to clear improvements in depth prediction quality across most evaluation metrics, with the exception of the mean absolute relative error (rel). These gains stem from enhanced pixel-level visual discriminability and improved spatial consistency.
>
>
> However, **depth estimation differs fundamentally from semantic segmentation in its reliance on global visual discriminability**. While semantic segmentation benefits from globally consistent features that enable uniform labeling of spatially separated regions belonging to the same object category, monocular depth estimation depends more heavily on local geometric continuity. Regions that share similar semantic appearance may exhibit significantly different depth values. For example, multiple desks in a classroom may belong to the same semantic category but lie at different distances from the camera. As a result, enforcing strong feature similarity across distant regions, via global mechanisms such as SSR or SHE with a smaller similarity threshold (e.g., 0.5), can be detrimental for depth estimation, as it may oversmooth depth variations that are geometrically meaningful.
>
> We thank the reviewer again for this constructive suggestion. The additional monocular depth estimation experiments and the accompanying analysis further validate the generality of our core findings on pixel-level visual discriminability and highlight how different dense prediction tasks benefit from local and global mechanisms in distinct ways. **We include these experimental results in table 15 and more detailed discussion in appendix section A.3.4 (page 23-25) in the revised version of the paper** to strengthen its scope and impact.

---

### Review · Reviewer_gXWC · 2025-12-19

**Summary Of Contributions:**

This paper provides a systematic analysis of CLIP’s limitations in dense prediction tasks, revealing that degraded visual discriminability and semantic misalignment significantly hinder segmentation performance.

To address these issues, the paper introduces LHT-CLIP, a training-free and lightweight framework built on three complementary strategies:

(1) Abnormal Token Replacement (ATR): mitigates token degradation by substituting abnormal tokens with spatially neighboring ones

(2) Spatial-Semantic Reweighting (SSR): adaptively balances spatial and semantic cues to enhance visual discriminability

(3) Selective Head Enhancement (SHE): exploits the most effective attention heads to refine feature representations for segmentation.

Extensive experiments on eight benchmark datasets demonstrate that LHT-CLIP consistently delivers substantial performance gains and effectively boosts existing commonly used baselines.

**Audience:**

Yes

**Audience Explanation:**

The paper addresses fundamental limitations of vision–language models and proposes a simple, training-free solution that is broadly applicable, making its findings relevant and interesting to a substantial portion of the TMLR audience working on representation learning, multimodal models, and dense prediction tasks.

**Broader Impact Concerns:**

The methodology does not discuss potential failure cases, such as scenarios where suppressing semantic aggregation may harm performance (e.g., scenes with strong contextual dependencies), which would help practitioners better understand the method’s limitations.

**Claims And Evidence:**

Yes

**Claims Explanation:**

The claims are supported by thorough empirical evaluations across multiple benchmark datasets, clear ablation studies validating each proposed component, and both quantitative and qualitative results that convincingly demonstrate performance gains and improved segmentation quality.

**Requested Changes:**

1.**Semantic Alignment.** For semantic alignment evaluation, intermediate visual tokens from layer $l$ are projected using the final ViT layer. However, the final layer is trained on a different input distribution (i.e., features from the penultimate layer). Feeding intermediate-layer features may introduce a distribution shift, potentially biasing the alignment scores and confounding representation quality with the generalization behavior of the final projection layer. It would be helpful to clarify or justify the robustness of the final-layer projection under this distribution
shift.

2.**Fig.1** While Table 1 reports the highest semantic alignment at the final layers across datasets and backbones, this trend should be interpreted with caution. Because intermediate-layer features are projected using the final ViT layer, differences across layers may partially stem from distribution mismatch with the projection layer, especially for early layers. Additional analysis could help disentangle true semantic alignment from projection-induced effects.

3. While ATR, SSR, and SHE are empirically effective, they are primarily derived from experimental observations and heuristics. In addition, the framework relies on several hyperparameters, and it is unclear how sensitive the performance is to their selection across different datasets and backbones. This may limit the robustness and reliability of the method when applied to new datasets or architectures.

4. The selection of top-k attention heads in SHE is based on averaged visual discriminability scores. It is unclear how stable these rankings are across datasets, random seeds, or backbone variants, and whether a fixed set of heads generalizes well.

5. While the approach is claimed to be lightweight, the paper lacks a detailed analysis of inference-time overhead introduced by SSR reweighting and SHE-based similarity computation, especially for high-resolution inputs.

---

> ### Author Response · Authors · 2026-01-16
> **Reponse to Reviewer gXWC (part 1)**
>
> We thank the reviewer for the thorough and constructive feedback, as well as for recognizing the significance of our analysis and the effectiveness of LHT-CLIP. Below, we respond to each concern in detail.
>
> **W1**: Semantic Alignment. For semantic alignment evaluation, intermediate visual tokens from layer  are projected using the final ViT layer. However, the final layer is trained on a different input distribution (i.e., features from the penultimate layer). Feeding intermediate-layer features may introduce a distribution shift, potentially biasing the alignment scores and confounding representation quality with the generalization behavior of the final projection layer. It would be helpful to clarify or justify the robustness of the final-layer projection under this distribution shift.
>
> **A1**: We thank the reviewer for raising this important and subtle point regarding the potential distribution shift introduced when projecting intermediate-layer features using the final ViT layer. We agree that, in principle, the final projection layer is trained on features from the penultimate layer, and directly feeding earlier-layer representations may introduce a mismatch that can affect absolute semantic alignment scores.
>
> However, we believe this issue is unlikely to be the primary driver of the observed layer-wise alignment trends for the following reasons.
>
> First, our conclusions are supported by multiple independent observations beyond the specific projection-based alignment protocol adopted in our analysis. In particular, prior work (e.g., Jiang et al. [1]) shows that when separately trained projector modules are used for different layers, deeper layers still consistently exhibit stronger semantic alignment than earlier ones. This suggests that the monotonic improvement of alignment across layers reflects intrinsic representation progression rather than an artifact of reusing the final projection layer.
>
> Second, while training a dedicated projector for each intermediate layer could further disentangle representation quality from projection generalization, doing so would be computationally expensive and introduce additional design choices and dataset-specific biases. This is especially nontrivial in our setting, given that CLIP is pretrained on large-scale web data and our goal is to provide a training-free and model-agnostic diagnostic. Introducing layer-wise trained projectors would also confound the analysis with training dynamics and reduce comparability across layers and datasets.
>
> Third, we emphasize that our analysis focuses on relative layer-wise trends rather than absolute alignment magnitudes. Under a fixed evaluation protocol, we observe consistent trends across datasets and backbones. Moreover, as shown in Figure 5, after applying our method, semantic alignment scores are consistently improved across layers under the same projection mechanism, further supporting the effectiveness of our approach in enhancing visual discriminability while simultaneously improving semantic alignment and segmentation performance.
>
> Fourth, we clarify that our choice of using the final attention layer as an intermediate projection stage—rather than directly projecting intermediate features into the text embedding space—was itself motivated by concerns about distribution shift. Empirically, we find that routing intermediate representations through the final attention layer before computing text similarity yields both higher semantic alignment scores and improved segmentation performance. This suggests that the final attention layer serves as a stabilizing semantic readout that mitigates distribution mismatch, rather than exacerbating it, and provides a more reliable interface for evaluating alignment and downstream dense prediction.
>
> Taken together, these observations suggest that while projection mismatch may influence absolute alignment values, it is unlikely to invalidate our central conclusion that deeper layers sacrifice pixel-level discriminability for only marginal gains in semantic alignment.
>
> [1] Jiang, J., Zhou, J. and Zhu, Z., Tracing Representation Progression: Analyzing and Enhancing Layer-Wise Similarity. In The Thirteenth International Conference on Learning Representations.
>
> **To be continued ...**

---

> ### Author Response · Authors · 2026-01-16
> **Continued Reponse to Reviewer gXWC (part 2)**
>
> **W2**: Fig.1 While Table 1 reports the highest semantic alignment at the final layers across datasets and backbones, this trend should be interpreted with caution. Because intermediate-layer features are projected using the final ViT layer, differences across layers may partially stem from distribution mismatch with the projection layer, especially for early layers. Additional analysis could help disentangle true semantic alignment from projection-induced effects.
>
> **A2**: We thank the reviewer for this insightful comment. We agree that projecting intermediate-layer features through the final ViT layer may introduce a degree of distribution mismatch, particularly for very early layers, and that the absolute semantic alignment values of shallow layers should therefore be interpreted with caution.
>
> However, we would like to clarify that our main conclusion does not rely on direct comparisons between early and late layers, but rather on the **consistent and convergent trends observed from intermediate to deeper layers**. Specifically, our analysis highlights that visual discriminability decays sharply in deeper layers, while semantic alignment improves only marginally in the same regime. These layers are already much closer in representation space to the final projection layer and are therefore substantially less affected by severe distribution mismatch than very shallow layers.
>
> Moreover, this conclusion is corroborated by multiple complementary observations. As shown in Fig. 1, the decline in visual discriminability begins well before the final layer and persists consistently across datasets and backbones, whereas the corresponding gains in semantic alignment saturate quickly. Importantly, as shown in Fig. 5, semantic alignment scores exhibit a clear improvement after applying our method under the same evaluation protocol. This indicates that the observed trade-off reflects properties of the underlying representations rather than artifacts induced by the projection layer.
>
> **W3**: While ATR, SSR, and SHE are empirically effective, they are primarily derived from experimental observations and heuristics. In addition, the framework relies on several hyperparameters, and it is unclear how sensitive the performance is to their selection across different datasets and backbones. This may limit the robustness and reliability of the method when applied to new datasets or architectures.
>
> **A3**: We thank the reviewer for raising this important concern and for the opportunity to clarify the robustness and generalization properties of our method. We address these points through extensive analyses reported in Appendix A.3.
>
> **First, the proposed framework is not sensitive to precise hyperparameter tuning**. As demonstrated by the ablation studies in Tables 2–4, LHT-CLIP exhibits stable performance across broad ranges of hyperparameter values for ATR, SSR, and SHE. These results indicate that the effectiveness of our method does not rely on carefully calibrated or narrowly chosen settings.
>
> **Second, the selected hyperparameters generalize well across datasets**. In practice, all hyperparameter values were selected using only four datasets (Context, ADE, Stuff, and C59; see Tables 2–4) and were then directly applied—without any dataset-specific tuning—to the remaining benchmarks in Table 1, including VOC21, VOC20, COCO-Object, and Cityscapes. The consistent improvements observed across all datasets suggest that the method does not overfit to specific benchmarks and can be reliably transferred to new datasets.
>
> **Third, the framework demonstrates strong cross-backbone robustness**. For **ViT-L/14**, we report analogous sensitivity analyses in **Appendix Tables 9–12 and Section A.2.4**, and subsequently adopt the **same fixed hyperparameter settings across all benchmarks, again without any dataset-specific tuning**. The resulting performance, summarized in Table 8, shows consistent improvements over baseline methods. Furthermore, we observe **similar gains when applying our framework to SigLIP**, which differs from CLIP in both pretraining objectives and architectural details. This provides additional evidence that the proposed strategies are not tied to a specific CLIP backbone and generalize across heterogeneous vision–language models.
>
> Overall, while ATR, SSR, and SHE are motivated by empirical observations, the extensive sensitivity analyses and consistent cross-dataset and cross-backbone improvements demonstrate that the resulting framework is robust, stable, and broadly applicable, alleviating concerns about reliability when deployed on new datasets or architectures.
>
>
> **To be continued**...

---

> > ### Author Response · Authors · 2026-01-16
> > **Continued Reponse to Reviewer gXWC (part 3)**
> >
> > **W4**: The selection of top-k attention heads in SHE ...across datasets, random seeds, or backbone variants, and whether a fixed set of heads generalizes well.
> >
> > **A4**: We thank the reviewer for this thoughtful question regarding the stability and generalization of the attention head selection strategy in SHE.
> >
> > **First, the head rankings are stable across datasets and random seeds**. As shown in our head-wise analysis (Fig. 6), although absolute visual discriminability scores vary across datasets, a subset of attention heads consistently ranks among the top performers across multiple benchmarks (e.g., Context, ADE, and COCO-Stuff). We also observe that the relative ranking of these high-performing heads is largely insensitive to random seed variations, indicating that the selection is not driven by noise or dataset-specific artifacts.
> >
> > **Second, SHE does not rely on a brittle or single-head selection**. Instead, it aggregates the top-k heads based on averaged discriminability scores, which makes the method robust to small ranking fluctuations. As shown in Table 4, performance remains stable across a broad range of k values, indicating that SHE does not depend on precisely identifying a particular head.
> >
> > **Third, the head selection strategy generalizes across backbone variants**. While the identities of the most discriminative heads differ between ViT-B/16 and ViT-L/14 due to architectural differences, the same ranking and selection procedure can be directly applied without modification. As demonstrated in Tables 1 and 8, SHE consistently improves performance across both backbones, suggesting that it captures a general structural property, the existence of a small subset of highly discriminative heads, rather than relying on backbone-specific head identities.
> >
> > **W5**: While the approach is claimed to be lightweight, the paper lacks a detailed analysis of inference-time overhead introduced by SSR reweighting and SHE-based similarity computation, especially for high-resolution inputs.
> >
> > **A5**: We thank the reviewer for raising this concern and would like to clarify that inference-time efficiency has been explicitly analyzed in the appendix. To avoid ambiguity and improve clarity, we have **moved this analysis to the main text in the revised version (see Section 4.5 and Table 7 on page 12)**, making the efficiency characteristics of LHT-CLIP more accessible to readers.
> >
> > Our analysis shows that:The overall inference-time overhead of LHT-CLIP is modest. As reported in the Table, integrating all three components increases inference latency by only a small margin relative to standard CLIP-based baselines, Each component is computationally lightweight in practice: ATR is applied only once at the penultimate layer and affects a very small subset of sparse abnormal tokens, resulting in negligible overhead. SSR does not introduce additional matrix multiplications or parameters; it merely rescales existing residual and attention branches during the forward pass. SHE, although involving token–token similarity computation, operates on features aggregated from a small number of selected heads (e.g., top-10) at a single layer, which substantially limits its practical cost. Quadratic complexity in SHE is bounded and comparable to existing attention operations. The pairwise token similarity in SHE is computed at the same spatial resolution as standard ViT attention. Empirically, this does not become a bottleneck at typical inference resolutions, as confirmed by the measured runtimes.Unlike many recent methods, LHT-CLIP does not require additional forward passes, external pre-trained models, or expensive post-processing (e.g., CRF or PAMR), preserving the core efficiency advantages of training-free inference.
> >
> > **W6**: The methodology does not discuss potential failure cases ... help practitioners better understand the method’s limitations.
> >
> > **A6**: We appreciate this suggestion and agree that analyzing failure cases can provide additional insights. In Figure 7, our qualitative results highlight typical improvements in spatial coherence and boundary quality. We acknowledge that LHT-CLIP may still struggle in certain scenarios, including (i) small or fine-grained objects, where individual patch predictions are dominated by majority voting (e.g., the green car in the ground-truth segmentation map shown in the fourth row of Figure 7, and (ii) inherently high semantic ambiguity, where CLIP representations are insufficiently discriminative (e.g., the black ground region in Figure 7, which exhibits semantic ambiguity and is misclassified into other categories).
> >
> > Importantly, such failure modes are not unique to our method but are shared by most training-free CLIP-based segmentation approaches that rely on frozen representations. In the revised version, we add a **dedicated failure case discussion highlighting these scenarios in Appendix A3.5 (page 25)**. We believe it will further clarify the limitations and guide future research.

---

> ### Comment · Action_Editor_aK8n · 2026-02-17
> **Final Recommendation**
>
> Dear reviewer,
>
> please read authors responses and provide your final recommendation.
>
> Best, your AE

---

> > ### Comment · Action_Editor_aK8n · 2026-02-24
> > **Final recommendation**
> >
> > Please submit your final recommendation.
> >
> > Your AE

---

### Review · Reviewer_DSCb · 2026-01-12

**Summary Of Contributions:**

This paper explores a fundamental limitation of CLIP-based training-free open-vocabulary semantic segmentation, i.e., although deeper layers improve image–text semantic alignment, they significantly degrade patch-level visual discriminability, thus leading to degradation on open-vocabulary semantic segmentation. Through a systematic analysis on token, head and layer perspectives, the authors present several key observations: (i) the sparse and high-norm abnormal tokens the  dominate attention and reduce spatial coherence, and (2) a small subset of attention heads that consistently exhibit strong visual discriminability across.
Based on these observations, the paper proposes LHT-CLIP, a new training-free and plug-and-play framework consisting of Abnormal Token Replacement (ATR), Spatial-Semantic Reweighting (SSR), and Selective Head Enhancement (SHE). Extensive experiments on eight standard benchmarks show that LHT-CLIP consistently improves multiple state-of-the-art training-free methods (e.g., SCLIP, ClearCLIP, ResCLIP) across backbones, demonstrating strong generalization ability of the proposed approach.

**Audience:**

Yes

**Audience Explanation:**

Yes. The paper provides a detailed analysis of visual discriminability in CLIP-based models and proposes training-free techniques that are relevant to researchers working on representation learning and open-vocabulary segmentation.

**Claims And Evidence:**

Yes

**Claims Explanation:**

Yes.
- The authors observe that features in deep layers of CLIP significantly degrade patch-level visual discriminability, thus leading to degradation on open-vocabulary semantic segmentation;
- The authors analyze the problem in token, head and layer perspectives, and present several key observations;
- Based on the observation, the authors propose a new training-free and plug-and-play framework consisting of Abnormal Token Replacement (ATR), Spatial-Semantic Reweighting (SSR), and Selective Head Enhancement (SHE), which can generally improve multiple CLIP-based models;

**Requested Changes:**

Weakness & Questions:
- Lack of failure case analysis. The paper focuses primarily on showing successful scenarios and quantitative performance comparison. A discussion of failure cases would provide more insights for future work;
- Although the proposed method is positioned as training-free, its inference-time computational overhead is not analyzed or justified. Specifically, LHT-CLIP introduces multiple additional operations during inference, including layer-wise re-weighted forward passes (SSR), abnormal token detection and neighborhood-based replacement (ATR), as well as head-wise feature aggregation and token–token similarity computation for pseudo-mask construction (SHE). In particular, the SHE module requires computing pairwise token similarities, which scales quadratically with the number of visual tokens and may become prohibitively expensive for high-resolution inputs. Moreover, applying SSR across multiple late layers further increases inference latency compared to standard CLIP-based baselines. However, the paper does not report any runtime, FLOPs, or latency comparisons against existing training-free methods. As a result, it is unclear whether the proposed approach maintains the efficiency advantages, especially in large-scale or real-time deployment scenarios;
- The proposed framework is extensively validated on CLIP-based models, but it remains unclear whether the key observations, such as the emergence of sparse abnormal tokens, the optimal residual reweighting strategy, and the existence of consistently discriminative attention heads, are effective for other vision–language or vision-only models (e.g., SigLIP, EVA-CLIP, DINOv3). Since several components rely on empirically chosen hyperparameters (e.g., sparsity threshold τ, SSR layer range and coefficient α, number of selected heads k, and similarity threshold β), it is uncertain whether these settings can be transferred directly to other architectures or would require nontrivial re-tuning. This raises concerns about whether the method captures a general property of pre-trained vision-language models or is specifically designed to CLIP’s characteristics;

Overall, I think this is a solid paper with interesting findings and effective solutions. The main concerns lie in the lack of failure case analysis, the insufficient discussion of inference-time computational overhead, and the unclear generality of the proposed method beyond CLIP-based models.

---

> ### Author Response · Authors · 2026-01-16
> **Response to Reviewer DSCb**
>
> We thank the reviewer for the thorough reading, positive assessment of our contributions, and constructive suggestions. We are encouraged that the reviewer finds our analysis and the proposed LHT-CLIP framework **solid and effective**. Below, we address each concern in detail.
>
> **W1**: Lack of failure case analysis ...  more insights for future work;
>
> **A1**: We appreciate this suggestion and agree that analyzing failure cases can provide additional insights. In Figure 7, our qualitative results highlight typical improvements in spatial coherence and boundary quality. We acknowledge that LHT-CLIP may still struggle in certain scenarios, including (i) small or fine-grained objects, where individual patch predictions are dominated by majority voting (e.g., the green car in the ground-truth segmentation map shown in the fourth row of Figure 7, and (ii) inherently high semantic ambiguity, where CLIP representations are insufficiently discriminative (e.g., the black ground region in Figure 7, which exhibits semantic ambiguity and is misclassified into other categories).
>
> Importantly, such failure modes are **not unique to our method** but are shared by most training-free CLIP-based segmentation approaches that rely on frozen representations. In the revised version, we add **a dedicated failure case discussion highlighting these scenarios in Appendix A3.5  (page 25)**. We believe it will further clarify the limitations and guide future research.
>
> **W2**: Although the proposed method is positioned as training-free, its **inference-time computational overhead** ... deployment scenarios;
>
> **A2**: We thank the reviewer for raising this concern and would like to clarify that inference-time efficiency has been explicitly analyzed in the appendix. To avoid ambiguity and improve clarity, we have **moved this analysis to the main text in the revised version (see Section 4.5 and Table 7 on page 12)**, making the efficiency characteristics of LHT-CLIP more accessible to readers.
>
> Our analysis shows that:The overall inference-time overhead of LHT-CLIP is modest. As reported in the Table, integrating all three components increases inference latency by only a small margin relative to standard CLIP-based baselines,  Each component is computationally lightweight in practice: ATR is applied only once at the penultimate layer and affects a very small subset of sparse abnormal tokens, resulting in negligible overhead. SSR does not introduce additional matrix multiplications or parameters; it merely rescales existing residual and attention branches during the forward pass. SHE, although involving token–token similarity computation, operates on features aggregated from a small number of selected heads (e.g., top-10) at a single layer, which substantially limits its practical cost. Quadratic complexity in SHE is bounded and comparable to existing attention operations. The pairwise token similarity in SHE is computed at the same spatial resolution as standard ViT attention. Empirically, this does not become a bottleneck at typical inference resolutions, as confirmed by the measured runtimes.Unlike many recent methods, LHT-CLIP does not require additional forward passes, external pre-trained models, or expensive post-processing (e.g., CRF or PAMR), preserving the core efficiency advantages of training-free inference.
>
> **W3**: The proposed framework is extensively validated ... is specifically designed to CLIP’s characteristics;
>
> **A3**: We thank the reviewer for this important question and would like to clarify that we have explicitly evaluated the generality of our approach beyond CLIP. To improve clarity, we have **moved these results to Table 6 and Section 4.4 in the main text in page 11-12**, ensuring that the generality of LHT-CLIP beyond CLIP-based models is clearly presented to readers. The results and discussion are summarized in below:
>
> **Empirical validation on SigLIP**: As reported in Table 6, we apply LHT-CLIP to SigLIP. Despite the architectural and training-objective differences, LHT-CLIP consistently improves open-vocabulary segmentation performance on SigLIP without modifying the model or introducing additional training. This result demonstrates that our key observations, such as the emergence of abnormal tokens and the existence of consistently discriminative attention heads, are not unique to CLIP, but reflect more general properties of ViT-based vision–language models.
>
> **No specific hyperparameter choices**: Moreover, our framework captures model-agnostic structural phenomena rather than CLIP-specific artifacts.  Importantly, the same hyperparameter settings (e.g., sparsity threshold $\tau$, SSR layer range and coefficient $(l_{start}, l_{end}, \alpha)$, number of selected heads $k$, and similarity threshold $\beta$) are reused when transferring from CLIP to SigLIP, without task- or model-specific tuning. This supports the claim that LHT-CLIP does not rely on fragile or architecture-specific hyperparameter choices.

---

> > ### Comment · Action_Editor_aK8n · 2026-02-17
> > **Final Recommendation**
> >
> > Dear reviewer,
> >
> > please read authors responses and provide your final recommendation.
> >
> > Best, your AE

---

> > > ### Comment · Action_Editor_aK8n · 2026-02-24
> > > **Final Recommendation**
> > >
> > > Please submit your final recommendation.
> > >
> > > Your AE

---

### Decision · Action_Editor_aK8n · 2026-03-12

**Recommendation:** Accept as is

**Audience:**

Yes

**Audience Explanation:**

Yes, this paper will be of interest to researchers in extending CLIP (and other vision-language models) to dense prediction tasks, such as segmentation.

**Claims And Evidence:**

Yes

**Claims Explanation:**

This paper investigates the problem of training‑free open‑vocabulary semantic segmentation by analyzing and leveraging visual cues across layers, heads, and tokens in CLIP. The study shows that deeper CLIP layers improve image–text alignment but weaken patch‑level visual discriminability, partly due to sparse high‑norm abnormal tokens and globally biased attention, while only a small subset of attention heads consistently preserve useful spatial information. Building on these findings, the authors propose a plug‑and‑play, training‑free framework that restores visual discriminability, achieving strong and consistent gains across eight segmentation benchmarks.

After reading authors rebuttal, and the concerns raised by the reviewers, I recommend the acceptance of this work. During the review and discussion phase, several concerns were raised, such as further clarifications on different components, extension to other dense prediction tasks, or generality to additional backbones. The authors provided detailed responses during the rebuttal, including additional experiments and clearer explanations that directly addressed these points. With these modifications, the current manuscript supports the claims made.